# Universal Mini-Batch Consistency for Set Encoding Functions

## Abstract

Previous works have established solid foundations for neural set functions, complete with architectures which preserve the necessary properties for operating on sets, such as invariance to permutations of the set elements. Subsequent work has highlighted the utility of Mini-Batch Consistency (MBC), the ability to sequentially process any permutation of a set partition scheme (e.g. streaming chunks of data) while guaranteeing the same output as processing the whole set at once. Currently, there exists a division between MBC and non-MBC architectures. We propose a framework which converts an arbitrary non-MBC model to one which satisfies MBC. In doing so, we allow all set functions to universally be considered in an MBC setting (UMBC). Additionally, we explore a set-based Monte Carlo dropout strategy which applies dropout to entire set elements. We validate UMBC with theoretical proofs, unit tests, and also provide qualitative/quantitative experiments on Gaussian data, clean and corrupted point cloud classification, and amortized clustering on ImageNet. Additionally, we investigate the probabilistic calibration of set-functions under test-time distributional shifts. Our results demonstrate the utility of UMBC, and we further discover that our dropout strategy improves uncertainty calibration.

## 1 Introduction

Set encoding functions (Zaheer et al., 2017; Bruno et al., 2021; Lee et al., 2019; Kim, 2021) have become a broad research topic in recent publications. This popularity can be partly attributed to natural set structures in data such as point clouds or even datasets themselves. Given a set of cardinality $N$, one may desire to group the elements (clustering), identify them (classification), or find likely elements to complete the set (completion/extension). A key difference from vanilla neural networks, is that neural set functions must be able to handle dynamic set cardinalities for each input set. Additionally, sets are considered unordered, so the function must make consistent predictions for any permutation of set elements.

Deep Sets (Zaheer et al., 2017) is a canonical work providing an investigation of the requirements and proposal of valid neural set function architectures. Deep Sets utilizes traditional, permutation equivariant (Property 3.2) linear and convolutional neural network layers in conjunction with permutation invariant (Property 3.1) set-pooling functions (*e.g.* {min, max, sum, mean}) in order to satisfy the necessary conditions and perform inference on sets. The Set Transformer (Lee et al., 2019) utilizes powerful multi-headed *self-attention* (Vaswani et al., 2017) to construct multiple set-capable transformer blocks, as well as an attentive pooling function. Though powerful, these works never explicitly considered the case where it may be required to process a set in multiple partitions at test time, which can happen for a variety of reasons including device resource constraints, prohibitively large or even infinite test set sizes, and streaming data conditions.

The MBC property of set functions was identified by Bruno et al. (2021) who also proposed the Slot Set Encoder (SSE), a specific version of a *cross-attentive* pooling mechanism which satisfies MBC, guaranteeing it will produce a consistent output for all possible piecewise processing of set partitions. The introduction of the MBC property naturally leads to the rise of a new dimension in the taxonomy of set functions, namely those which satisfy MBC and those which do not. The SSE is an example of *one* valid MBC architecture which comes at the cost of eliminating powerful self-attentive models such as the Set Transformer. Self-attention can be the best choice for tasks which require leveraging

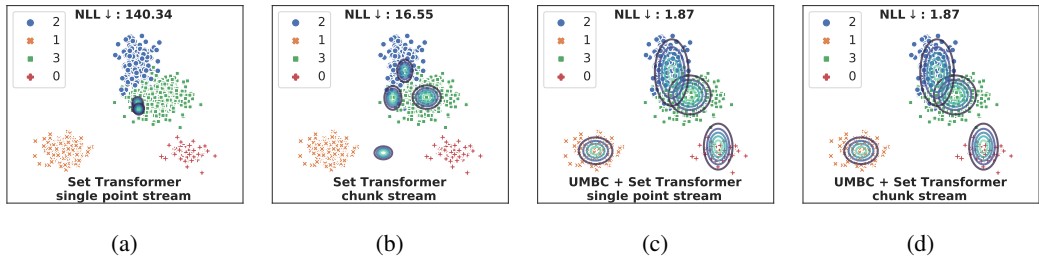

(a)  (b)  (c)  (d)

Figure 2: ($\circ$, $+$, $\times$, $\square$) correspond to classes in the input set. Ellipses are the model's clustering prediction. **In streaming settings, models must process the stream without storing streamed inputs. a-b**: The Set Transformer delivers poor likelihood on different set streams. **c-d**: Set Transformer with a UMBC module becomes an MBC function, yielding better likelihood, and consistent predictions regardless of the data stream. For a description of streaming settings, see Section 5; additional streams are shown in Figure 8.

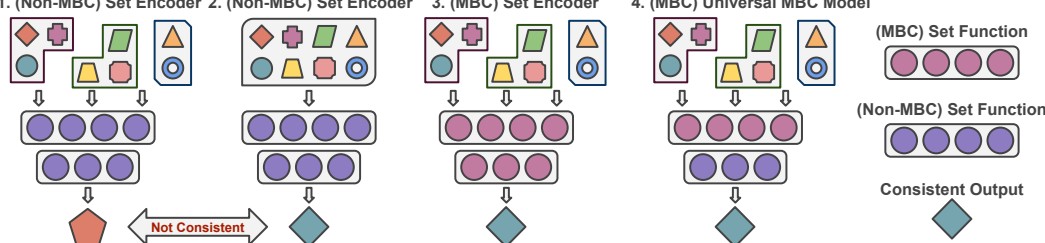

Figure 3: **Non-MBC** set functions (1 & 2) are not MBC when sequentially processing random set partitions. **MBC** set functions (3) are MBC, but with limited valid architectures. **Universal MBC** (4) (UMBC) allows leveraging MBC+non-MBC set functions, widening the field of available MBC architectures.

pairwise relationships between set elements such as clustering (as we show later in results in Figure 4 and Table 2) where the Set Transformer outperforms SSE).

Models such as the Set Transformer cannot make MBC guarantees when updating pooled set representations, as self-attention blocks require all $N$ elements in a single pass, and therefore do not satisfy MBC (*i.e.* processing separate pieces of a set yields a different output than processing the whole set at once). Naively using such non-MBC set functions in an MBC setting can cause a severe degradation in performance, as depicted in Figures 2a and 2b where the Set Transformer exhibits poor likelihood and inconsistent clustering predictions. With the addition of a UMBC module UMBC+Set Transformer inherits an MBC guarantee, yielding consistent results, and much higher likelihoods (Figures 2c and 2d) (See Section 5 and Appendix B for details of the experiment). The quantitative effect of MBC vs non-MBC encoding on a pooled set representation can be seen in Figure 1 which shows the variance between pooled representations of 100 random partitions of the same set. (See Appendix C for details).

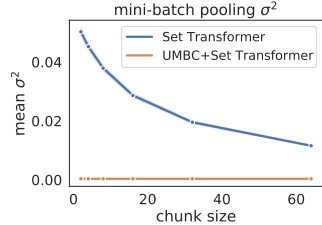

Figure 1: $\sigma^2$ between encoded features of 100 random mini-batched set partitions. Set Transformer (not MBC) produces variance in the output. UMBC+Set Transformer produces consistent output for all 100 random mini-batched partitions.

In this work, we propose, and verify both theoretically and empirically that there exists a universal method for converting arbitrary non-MBC set functions into MBC functions, providing MBC guarantees for mini-batch processing of random set partitions. This allows for any set encoder to be used in an MBC setting where it may have previously failed (*e.g.* streaming data). This result has large implications for all current and future set functions which are not natively MBC, which can now be used in a wider variety of settings and under more restrictive conditions. Animations, code, and tests can be found in the supplementary file and also at: https://github.com/anonymous-subm1t/umbc

**Our contributions in this work are as follows:**

- In Theorem 4.1 we show that any arbitrary non-MBC set encoder can become MBC, guaranteeing that mini-batch processing of sets of any cardinality at test-time will give the same result as processing the full set at once.

- We show that in tasks where non-MBC set functions outperform previous MBC functions, adding UMBC produces the strongest MBC model (Figure 4), highlighting the novelty and utility of UMBC.

- With Proposition 4.1, we loosen the constraints on the MBC attention activation functions proposed by (Bruno et al., 2021) (which only uses a sigmoid) by showing a wide variety of functions can be used (*e.g.* traditional attention softmax (Vaswani et al., 2017)).

- We explore an interesting MBC Monte Carlo dropout approach made possible by our UMBC module which delivers improvements in calibration for both in-distribution (Figure 7) and corrupted test sets (Figure 6) while still maintaining MBC guarantees.

## 2  RELATED WORK

Table 1: The MBC status of various set functions and the types of functions they contain.

| Model | MBC | Cross-Attn., | Self-Attn. |
|---|---|---|---|
| Deep Sets (Zaheer et al., 2017) | ✓ | ✗ | ✗ |
| SSE (Bruno et al., 2021) | ✓ | ✓ | ✗ |
| Set Transformer (Lee et al., 2019) | ✗ | ✓ | ✓ |
| UMBC+Set Transformer | ✓ | ✓ | ✓ |

Processing, pooling, and making a predictions for set structured data has been an active topic since the introduction of DeepSets (Zaheer et al., 2017). Attention has also been shown to be powerful in these tasks (Lee et al., 2019), as simple independent row-wise operations may fail to capture pairwise interactions between set elements. Subsequent works have drawn connections between set attention and optimal transport (Mialon et al., 2020), and subsequently expectation maximization (Kim, 2021). Likewise, efficient versions of set-attention have been proposed which incorporate cross attention with low dimensional self-attention in an iterative process (Jaegle et al., 2021). Outside of attention, other approaches to set pooling functions include featurewise sorting (Zhang et al., 2019), and canonical orderings (Murphy et al., 2018) to tackle the problems posed by the required permutation invariance.

Bruno et al. (2021), provide an especially important lens through which to view our work. Prior to the proposal of the MBC property, previous works never explicitly considered the MBC setting, which will likely become important with the ever increasing scales of models and data (Brown et al., 2020). Indeed most set functions do not satisfy Property 3.3 (*e.g.* (Lee et al., 2019; Kim, 2021; Mialon et al., 2020; Jaegle et al., 2021; Zhang et al., 2019; Murphy et al., 2018)). Our work builds on the concepts established by Bruno et al. (2021), and ensures that all set functions proposed in the future can be considered in MBC settings by incorporating UMBC.

Several prior works (Ovadia et al., 2019; Guo et al., 2017) highlight the problem of uncertainty quantification and probabilistic calibration, which can be crucial for tasks such as autonomous driving (Chen et al., 2017) and medical diagnoses (Zhou et al., 2021) where decisions can impact human well being. Guo et al. (2017), proposed quantifying uncertainty with the expected calibration error (ECE) metric measuring the mismatch between accuracy and confidence. Ovadia et al. (2019) used corrupted datasets such as ImageNet-C (Hendrycks & Dietterich, 2019) to survey the landscape of neural network calibration. We take a similar approach for set functions with ModelNet40-C (Ren et al., 2022) in our experiments. Guo et al. (2017); Ovadia et al. (2019) analyze calibration in variants of deep convolutional models, while Minderer et al. (2021) evaluate the calibration of large Vision Transformers. To our knowledge, our work is the first to analyze set function calibration specifically, as most other works focus on general purpose classifiers.

## 3  PRELIMINARIES ON SET FUNCTIONS

For our setting, we define a neural set function $f : \mathcal{X} \mapsto \mathcal{Y}$ with a set-structured input space $\mathcal{X}$ and output space $\mathcal{Y}$. $f$ operates on sets $X = \{\mathbf{x}_i\}_{i=1}^N$ with each set element $\mathbf{x}_i \in \mathbb{R}^d$. A dataset of set-structured data $\mathcal{D} = \{(X_i, Y_i)\}_{i=1}^M$ consists of input sets $X_i$ and output sets $Y_i$, the mapping of which can be learned by $f$ via stochastic gradient descent. Importantly, an input $X_i$ is a set and $f$ must process any set, therefore any element of the powerset $\mathscr{P}(X_i)$ also represents a possible input.

Deep Sets (Zaheer et al., 2017) provided a crucial groundwork for neural set functions, formalizing the requirements of permutation equivariant (Property 3.2) architectures and invariant (Property 3.1) pooling mechanisms necessary for feature extraction, pooling, and predictions on sets. Following

these requirements, a function can assign a (possibly different) output for each valid subset $X_j \in \mathscr{P}(X_i)$, although the prediction should be invariant to permutations of the elements within $X_j$.

**Property 3.1** (Permutation Invariance). *A function $f : \mathscr{P}(\mathcal{X}) \mapsto \mathcal{Y}$ acting on sets is permutation **invariant** to the order of objects in the set iff for any permutation function $\pi$ : $f(\mathbf{x}_1, \ldots, \mathbf{x}_n) = f(\mathbf{x}_{\pi(1)}, \ldots, \mathbf{x}_{\pi(N)})$*

Permutation invariant layers are commonly referred to as set pooling functions, and generally have a fixed size output given any permutation or cardinality of the input set. Such a permutation invariant function can also be called a Set to Vector function, as it produces a vector of a fixed size given a variable sized input.

**Definition 3.1** (Set 2 Vector Function). *A Set to Vector Function (S2V) $f : \{\mathbf{x}\} \mapsto \mathbb{R}^{d'}$ is a set function satisfying Property 3.1, which projects a set $\{\mathbf{x}_i \in \mathbb{R}^d\}_{i=1}^N$ to one or more vectors $\mathbf{z}_i \in \mathbb{R}^{d'}$.*

Additionally, Zaheer et al. (2017) prescribes that prior to any permutation invariant pooling, any composition of permutation equivariant layers may be used for feature extraction. Common linear and convolutional neural network layers are valid permutation equivariant functions when considering a batch of inputs as a set. For the remainder we assume $f$ contains both equivariant and invariant layers.

**Property 3.2** (Permutation Equivariance). *A function $f : \mathscr{P}(\mathcal{X}) \mapsto \mathcal{Y}$ acting on sets is permutation **equivariant** to the order of objects in the set iff for any permutation function $\pi$ : $\mathbf{f}([\mathbf{x}_{\pi(1)}, \ldots, \mathbf{x}_{\pi(N)}])^\top = [f_{\pi(1)}(\mathbf{x}_1), \ldots, f_{\pi(n)}(\mathbf{x}_N)]^\top$*

Lee et al. (2019) identified that self-attention layers (Vaswani et al., 2017) satisfy Property 3.2 and thus can be used as equivariant feature extractors for set functions, creating the Set Transformer. Set Attention Blocks (SAB) are defined as $\text{SAB}(X, X) = \text{Attention}(X, X) = \text{softmax}(X_q X_k^\top) X_v$ (*i.e.* Self-Attention (Vaswani et al., 2017)). Additionally, the permutation invariant pooling layer of the Set Transformer, Pooling by Multihead Attention (PMA), performs cross-attention between a learnable seed parameter $S \in \mathbb{R}^{K \times d}$ and the input set, $\text{PMA}(X) = \text{SAB}(S, X) \in \mathbb{R}^{K \times d}$.

Bruno et al. (2021) identified the MBC property, proposing the MBC Slot Set Encoder (SSE), adding a new dimension to Property 3.1 from Zaheer et al. (2017). Instead of merely requiring that $f(X)$ be invariant to permutations of the indices $i$ of $\mathbf{x}_i \in X$, the MBC property also requires that sequential processing of partitions, with partition indices $p$ for $\mathbf{x}_{i,p} \in X$ is also permutation invariant.

**Property 3.3** (Mini-Batch Consistency (MBC)). *Let $X = \{\mathbf{x}_i\}_{i=1}^N$ be a set with each element $\mathbf{x}_i \in \mathbb{R}^d$. Let $X$ be partitioned such that $X = \bigcup_{j=1}^{|P|} X_{P_j}$ and $f : \{\mathbf{x}\} \mapsto \mathbb{R}^{d'}$ be a S2V function. Given an aggregation function $g : \{f(X_j)\}_{j=1}^{|P|} \mapsto \mathbb{R}^{d'}$, $g$ and $f$ are Mini-Batch Consistent iff the following holds for any permutation of any random partition scheme,*

$$g\big(f(X_{P_1}), \ldots, f(X_{P_n})\big) = f(X)$$

The $f$ function of an SSE (Bruno et al., 2021) layer uses cross-attention with slots $S \in \mathbb{R}^{K \times d}$ as queries, and partitions $X_{j \in P}$ as keys and values. An SSE then utilizes an elementwise attention matrix activation which does not depend on the other $N - 1$ elements within the set. SSE proposes a sigmoid activation $\sigma$ in the attention matrix $A = \sigma(SX^\top)$ which is then normalized over the slot dimension $K$ (Bruno et al., 2021; Locatello et al., 2020) such that $A'_{i,j} = A_{i,j} / \sum_{i=1}^K A_{i,j}$. With $X_j \in \mathbb{R}^{|X_j| \times d}$, and $\hat{\sigma}$ being the slot-normalized sigmoid,

$$\text{Attention}^1(S, X) = \hat{\sigma}(SX^\top)X = \sum_{j=1}^P \hat{\sigma}(SX_j^\top)X_j, \tag{1}$$

where the sum is over the partition cells, thereby consistently guaranteeing the same output given any partition scheme and satisfying Property 3.3. The MBC aggregation function $g \in \{\min, \max, \text{mean}, \text{sum}\}$ (Bruno et al., 2021) is represented by the sum on the RHS of Equation (1).

---

[1]We omit the scaling of $QK^\top$ by $\frac{1}{\sqrt{d}}$ for brevity throughout this text, but include the scaling in our code.

# 4 BUILDING A UNIVERSALLY MBC SET FUNCTION

An important insight is that the SSE acts as a S2V function, creating an encoded set representation for subsequent processing. However, a S2V function which outputs $K$ vectors $\mathbf{z}_i \in \mathbb{R}^{d'}$, as stated in Definition 3.1 can be re-interpreted as outputting a set of cardinality $K$ with each element $\mathbf{z}_i \in \mathbb{R}^{d'}$. Therefore, by using a S2V function with $K > 1$ as the base module $f$, we can view the S2V function as a function which takes a set of cardinality $N$ and maps it to a set of cardinality $K$. Therefore, a downstream non-MBC module Property 3.3 need only be satisfied up until an invariant S2V pooling function.

**Theorem 4.1.** *Let $f^*$ be a set function satisfying either Property 3.2 or Property 3.1, and $(g, f)$ be functions satisfying Property 3.3, which together form the functional composition $F = f^* \circ g \circ f$. For $F$ to satisfy Property 3.3, it is sufficient to require the representation $Z = g(f(X_1), \ldots, f(X_p))$ as input to $f^*$ to satisfy Property 3.3.*

*Proof.* Assume that $g \circ f$ satisfies Property 3.3 and the composition $F$ does not satisfy Property 3.3. $g \circ f$ updates $Z$ as new partitions $X_j$ arrive, yielding an MBC input to $f^*$, and therefore the same output of $F$ for any permutation of a random partition of $X$, contradicting the statement that $F$ does not satisfy Property 3.3. □

Put simply, Theorem 4.1 states that every module $f^*$ coming after a module which satisfies Property 3.3 will continue to satisfy Property 3.3, even though $f^*$ itself may not satisfy Property 3.3. With this established, we can therefore use Theorem 4.1 to build a universally MBC set function.

---

**Algorithm 1** Universal MBC Module with set function $f^*$ and slots $S \in \mathbb{R}^{K \times d'}$.

---

1: **Input:** Partitioned Set $X = \{X_1, \ldots, X_p\}$
2: **Output:** $\hat{S} \in \mathbb{R}^{\hat{K} \times d'}$
3: **Set:** $Z \leftarrow 0$
4: **for** $i = 1$ **to** $p$ **do**
5:     // to satisfy MBC, $g=+$ and $f$=Attention
6:     $Z \leftarrow Z + \text{Attention}(S, X_i)$ (Equation (3))
7: **end for**
8: **return** $f^*(Z)$

---

**Maintaining attention normalization over $N$**
We now turn to the question of whether or not the constrained attention operation (*i.e.* avoiding normalization over $N$ in the attention activation) described for SSE is in fact necessary in order to satisfy Property 3.3.

**Proposition 4.1.** *By factorizing a normalization constant which depends on all $N$ set elements from the attention matrix[2], normalization over $N$ can be performed across mini-batched partitions while still satisfying Property 3.3.*

*Proof.* With $\phi$ as an elementwise activation function applied to the attention matrix, and $\hat{\phi}$ representing the same function with normalization over $N$ elements (*i.e.* like the traditional softmax in dot product attention),

$$\text{Attention}(S, X) = \hat{\phi}(SX^\top)X = \text{diag}(\boldsymbol{\zeta})^{-1}\phi(SX^\top)X \tag{2}$$

Where $\text{diag}(\boldsymbol{\zeta})$ is a diagonal matrix containing the normalization constants $\zeta_k = \sum_{i=1}^{N} \phi(\mathbf{x}_i^\top \mathbf{s}_k)$ where $N$ is the set cardinality and $\mathbf{s}_k$ is a single slot. Outside $\phi(.)$, the final multiplication is associative, so we may simply evaluate $\phi(SX^\top)X$, keeping a vector $\boldsymbol{\zeta}$ with the incrementally updated normalization constants (*e.g.* the sum of the rows of $\exp(SX^\top)$ for a softmax function). Factoring the attention in this way, we can update $\boldsymbol{\zeta}$ and $\phi(SX^\top)X$ at the arrival of every partition $X_j$, normalize over $N$, and still satisfy Property 3.3.

$$\text{Attention}(S, X) = \text{diag}(\boldsymbol{\zeta})^{-1}\exp(SX^\top)X = \text{diag}\Big(\sum_{j=1}^{P}\boldsymbol{\zeta}_j\Big)^{-1}\sum_{j=1}^{P}\exp(SX_j^\top)X_j \tag{3}$$

□

Interestingly, in our ablation study (Figure 5), we find the softmax most effective, which requires the normalization over $N$ as described above. For a note about about the numerical stability of the softmax calculated this way, see Appendix F.

---

[2]We consider the dot product attention kernel, but any valid attention kernel function may be used.

**Monte Carlo Slot Dropout** As outlined in Section 4, our UMBC framework projects a set of cardinality $N$ to a fixed cardinality $K$. In doing so, there is a unique opportunity where we can treat each slot index $\{i\}_{i=1}^{K}$ as a Bernoulli random variable, dropping it with probability $p$ (*i.e.* dropout (Hinton et al., 2012; Gal & Ghahramani, 2016) on the $K$ set elements). This can be done while still satisfying Property 3.3, as any dropout noise would be placed after the S2V function and thus be considered part of $f^*$. Uniquely, a UMBC module can then perform Monte Carlo (MC) dropout (Gal & Ghahramani, 2016) on a streaming set, while never seeing the entire set at once. Dropping slot indices strictly decreases the cardinality of the set which is input to $f^*$. During training, this strategy could be useful for faster training, due to reduced set size for $f^*$ (see Figure 18 for example), for regularization (Table 8 and Appendix J), or achieving a test-time ensemble of predictions by MC integration (Figure 7) as done by Gal & Ghahramani (2016). Specifically, for test time MC integration, considering $\mathbf{m}$ as a binary vector which selects the indices of set elements for processing by $f^*$, that is, $f^*(f(X), \mathbf{m}) = f^*(\{\mathbf{z}_i \in f(X) : m_i = 1\})$, we perform the following approximation with dropout rate parameter $p$ and sample size $S$.

$$p(y|X) = \int p(y|X, \mathbf{m})p(\mathbf{m})d\mathbf{m} \approx \frac{1}{S}\sum_{i=1}^{S} f^*(f(X), \mathbf{m}), \quad m_i \sim \text{Bernoulli}(p) \tag{4}$$

**Parallel UMBC Heads.** We may also consider $L$ multiple parallel UMBC set projections layers, with each layer $f_i$ projecting the input of cardinality $N$ to cardinality $K$ to form an input to $f^*$ of cardinality $LK$. Specifically $f^*(\bigcup_{i=1}^{L} f_i(X))$. Each $f_i$ would have independent multiheaded attention, allowing for independent representations of the same input set. Theorem 4.1 requires projecting a set of cardinality $N$ to a set of cardinality $K$, which *may* be seen as a bottleneck. Therefore parallel UMBC blocks could be used to add more expressiveness and reduce the overall effect of the UMBC bottleneck.

## 5 EXPERIMENTS

**Metrics & Setup** In the streaming settings of Figures 2 and 8 we placed arbitrarily hard MBC constraints (*i.e.* streaming settings described in Appendix B.1) to show how failure can occur in a non-MBC model. However, it is unclear how to attempt to fairly compare against non-MBC models on MBC tasks. Additionally, the UMBC property has been proven both theoretically (Theorem 4.1) and empirically (Figures 1, 2 and 8) Thus, in the following experiments, our aim is to evaluate the overall effect of the composition $f^*(\text{UMBC}(.))$, with the understanding that MBC models such as DeepSets and SSE extend to MBC settings while non-MBC models could be forced to perform in arbitrarily extreme MBC settings (like a single point stream).

We perform an ablation study on the components of UMBC, amortized clustering on Mixtures of Gaussians (MoG) and ImageNet-1K, as well as ModelNet40 point cloud classification. Additionally, we analyze the calibration of popular set-functions on ModelNet40-C, which has not been examined in any prior work to our knowledge. We report accuracy, negative log likelihood (NLL), expected calibration error (ECE) (Guo et al., 2017), and Adjusted Rand Index (ARI) (Hubert & Arabie, 1985; Vinh et al., 2010). Standard settings of all UMBC models follow those shown in Table 6 unless otherwise specified. All models are trained over 5 different random initializations, with one standard deviation error bars. We use open source code baseline code by Zaheer et al. (2017); Lee et al. (2019); Kim (2021) where applicable.

**Amortized clustering** We consider amortized clustering on a similar Gaussian dataset as Lee et al. (2019) (See Appendix B for dataset details). Figures 2 and 8 contain a qualitative example of the task as well as a demonstration of how non-MBC models fail when used in an MBC setting. The task objective is to maximize the likelihood (Equation (5)) of a set with $K$ Gaussian components by predicting the component prior, mean, and covariance of each mixture component $f(X) = \{\pi(X), \{\mu_j(X), \Sigma_j(X)\}_{j=1}^{K}\}$,

$$\log p(X; \theta) = \sum_{i=1}^{N} \log \sum_{j=1}^{K} \pi_j \mathcal{N}(x_i; \mu_j, \Sigma_j) \tag{5}$$

We show the effect train/test set sizes in the full batch setting in Figure 4. Interestingly, the best performing MBC models (Figures 4e and 4f) are UMBC+(Diff. EM, Set Transformer), Intuitively, this happens because the other MBC models (DeepSets and SSE) are not able to leverage pairwise

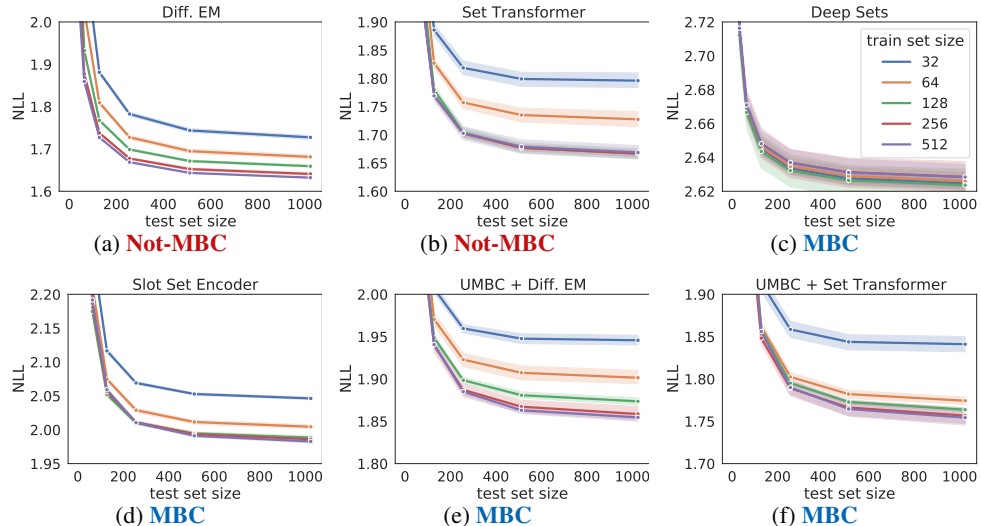

Figure 4: **Among models which satisfy Property 3.3 (c,d,e,f) UMBC+Set Transformer Shows the top performance. c-b** show unmodified functions. **e and f** show UMBC augmented functions. UMBC+Set Transformer allows the power of a transformer to be used in an MBC setting. The legend in the top left figure applies to all plots. This experiment showcases the same task as seen in Figure 2

relationships between set elements. In UMBC, everything after the first function $f$ can be arbitrarily complex, allowing $f^*$ to leverage self-attention. Crucially, this task requires a global view of relationships between points in order to predict cluster parameters. Therefore, it is intuitive to see why $f^*$'s utilizing self-attention or expectation maximization perform better than simple $f^*$'s (such as row-wise linear layers in Deepsets). Note that this is the same task as depicted in Figures 2 and 8, which shows how the better bottom line performance of the vanilla Set Transformer in Figure 4 evaporates in the MBC setting.

We extended the amortized clustering to ImageNet-1K (Deng et al., 2009), using features extracted from a pretrained ResNet50 (He et al., 2016) model (See details in Appendix E). Results are shown in Table 2. The Empirical model in Table 2 is the NLL and ARI obtained using the actual prior, empirical mean, and diagonal covariance of each class cluster. In this task, UMBC+Set Transformer outperforms all other

Table 2: Amortized Clustering on ImageNet features extracted with a pre-trained ResNet50.

| Model | MBC | NLL ↓ | ARI ↑ |
|---|---|---|---|
| Empirical | - | 1028.22±1.24 | 44.09±0.11 |
| Deep Sets (Zaheer et al., 2017) | ✓ | 531.44±0.15 | 6.18±0.08 |
| SSE (Bruno et al., 2021) | ✓ | 520.29±0.63 | 22.91±1.85 |
| Diff. EM[3](Kim, 2021) | ✗ | 524.74±0.38 | 13.22±0.16 |
| Set Transformer (Lee et al., 2019) | ✗ | 512.59±0.33 | 17.13±3.67 |
| UMBC+Diff. EM | ✓ | 518.56±0.92 | 13.04±0.45 |
| UMBC+Set Transformer | ✓ | **503.89±0.87** | **23.68±1.85** |

models. To account for UMBC's added parameters, we included UMBC on the baseline MBC models in Table 7, and UMBC+Set Transformer still shows the top performance. These results show that as UMBC *may* act as a bottleneck, there is no guarantee that the bottleneck will always negatively effect the performance of the underlying model.

**SSE's Connection to PMA's** With the introduction of Proposition 4.1, it is easy to see that the only difference between an SSE and a PMA is the choice of the attention activation function. Indeed any deterministic elementwise function which **1)** maps the pre-activation attention matrix to strictly positive values, and **2)** has an

Table 3: Valid UMBC attention activation functions. $K$ norm. and $N$ norm. refer to the the normalization constant over the slots $K$ and instances $N$[4], respectively.

| function ($\sigma$) | $K$ norm. | $N$ norm. | name | reference |
|---|---|---|---|---|
| $\text{sigmoid}(x_i)$ | sum | - | slot-sigmoid | (Bruno et al., 2021) |
| $\exp(x_i)$ | sum | sum | slot-softmax | (Locatello et al., 2020) |
| $\exp(x_i)$ | - | sum | softmax | (Lee et al., 2019) |
| $\exp(x_i)$ | $\mathbf{x} - \max_i(\mathbf{x})$ | sum | slot-exp | - |
| $\text{sigmoid}(x_i)$ | - | sum | sigmoid | - |

---

[3]Diff. EM showed some instability on the ImageNet clustering task and failed to converge for one run. Therefore variance is reported on 4/5 runs.

[4]In cases where there is both a $K$ norm and an $N$ norm, the $K$ norm is performed first.

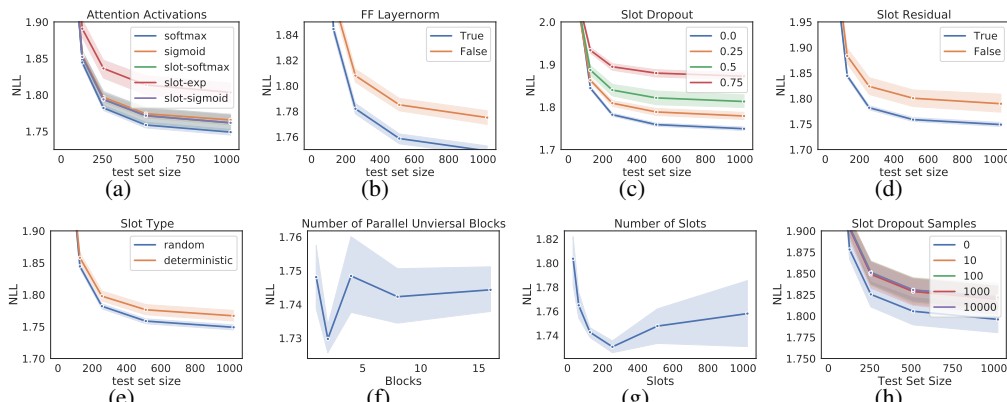

Figure 5: Ablation study, analyzing the effects of different settings within the UMBC module. These experiments were performed on the MoG dataset with the UMBC+Set Transformer model.

Table 4: Point cloud classification (ModelNet40). Models are trained on a set size of 1000 randomly sampled points, and evaluated on 100, 1000, and 2048 (max) test set sizes. **bold** entries denote the best performance between models with and without UMBC. Models in the top row are for reference.

| Model | MBC | Accuracy ↑ 100 | 1000 | 2048 | NLL ↓ 100 | 1000 | 2048 | ECE ↓ 100 | 1000 | 2048 |
|---|---|---|---|---|---|---|---|---|---|---|
| Deep Sets (Zaheer et al., 2017) | ✓ | 65.37±1.07 | 88.35±0.32 | 88.72±0.21 | 1.57±0.03 | 0.40±0.01 | 0.40±0.01 | 17.38±0.95 | 4.21±0.27 | 4.02±0.16 |
| SSE (Bruno et al., 2021) | ✓ | 71.09±0.51 | 87.85±0.39 | 87.92±0.42 | 1.42±0.10 | 0.52±0.05 | 0.51±0.06 | 16.69±1.11 | 5.93±1.06 | 5.88±1.17 |
| Diff-EM (Kim, 2021) | ✗ | 62.67±1.21 | 86.08±0.12 | **86.86±0.36** | 2.40±0.11 | 0.71±0.02 | 0.69±0.03 | 22.16±0.93 | 5.15±0.11 | 4.96±0.28 |
| UMBC+Diff-EM | ✓ | **67.07±1.67** | **86.22±1.23** | 86.37±1.03 | **1.61±0.12** | **0.58±0.06** | **0.57±0.05** | **13.97±1.51** | **4.32±1.37** | **4.38±1.27** |
| Set Transformer (Lee et al., 2019) | ✗ | **74.21±1.67** | **87.81±0.44** | **88.17±0.32** | 1.76±0.08 | 0.79±0.08 | 0.78±0.08 | 17.12±0.46 | 7.48±0.62 | 7.37±0.54 |
| UMBC+Set Transformer | ✓ | 71.18±1.52 | 86.56±0.49 | 86.77±0.29 | **1.23±0.15** | **0.53±0.03** | **0.51±0.03** | **10.37±2.24** | **2.60±0.19** | **2.35±0.24** |

optional normalization constant over $N$ which can be factored as in Proposition 4.1 is valid and will satisfy Property 3.3. With this in mind, we identify five functions, and explore their effects in Table 3 and Figure 5.

**Ablation Study**    Using the mixture of Gaussians dataset for amortized clustering, we analyze components of UMBC+Set Transformer in Figure 5. Of the five activation functions identified as valid in Section 4 we found that the softmax used in traditional attention performs the best. In agreement with Bruno et al. (2021), we find that treating the slots as a Gaussian random variable, and learning them with reparameterization (outlined in Appendix H) leads to better overall results. We also find that layernorm on the post attention linear layer, residual connections on the slots before the FF layer (like the PMA of Lee et al. (2019)) to be beneficial. Increasing the number of slots (cardinality of input to $f^*$), helped to a point and then showed a decrease in performance, likely due to overparameterization. We used these settings to inform our base settings given in Table 6.

The effect of slot dropout at both train time and test time can be seen in Figure 5 (c, and h). Empirically, on the Gaussian clustering task, we found that using no slot dropout ultimately led to the best performance, which we think is likely due to the fact that the dataset can sample infinitely many instances and is therefore extremely resistant to overfitting. Using dropout on the ModelNet40 dataset (Figure 7), which is prone to overfitting, led to increased performance on all metrics.

**Point cloud classification**    We perform set classification experiments ModelNet40 (Wu et al., 2015) and analyze the robustness of different set encoders to dataset shifts and varying test-time set sizes using ModelNet40-C (Ren et al., 2022) which contains 15 corruptions at 5 levels of intensity. Our experiments use the version of ModelNet40 and ModelNet40-C used by Ren et al. (2022) which contains 2048 points sampled from the original ModelNet40 (Wu et al., 2015) CAD models. Results are presented in Table 4 and Figure 6. For non-MBC models, augmenting the model with UMBC only shows a performance decrease for the larger set sizes, likely due to the bottleneck caused by the projection from cardinality $N$ to $K$. Interestingly, on set sizes smaller than the model was trained on, UMBC augmented models show consistently higher accuracy and indicates that UMBC is resilient to smaller test set sizes. In terms of ECE and NLL, UMBC models outperform all non-MBC baselines.

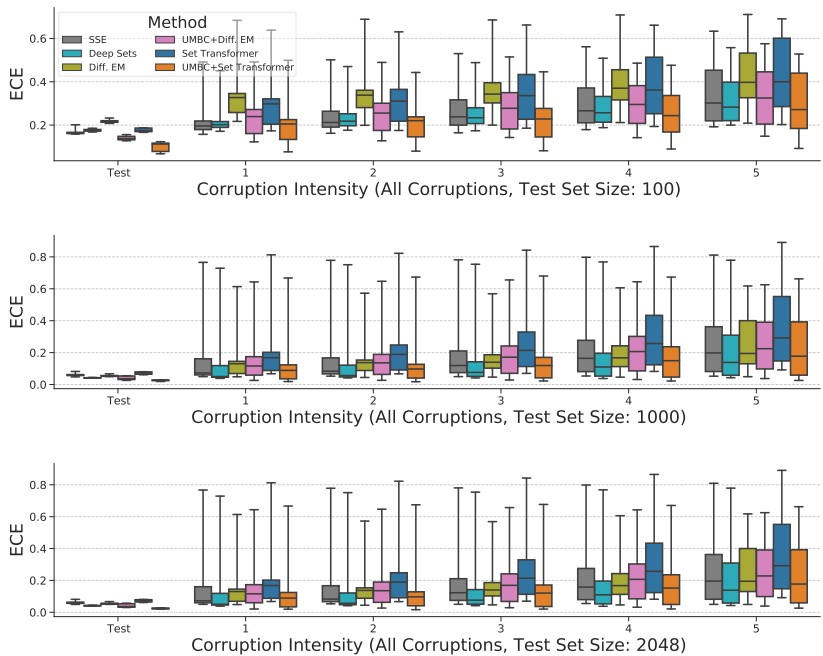

Figure 6: Expected Calibration Error on ModelNet40-C which contains 15 corruptions at 5 different intensity levels. 'Test' corresponds to the uncorrupted test set. See Figures 13 and 14 for Accuracy and NLL and Figures 15 to 17 for results on individual corruptions.

This increase in ECE can be partly attributed to MC sampling slots at test time (shown in Figures 7, 11 and 12) and partly to Slot Dropout at train time (shown in Table 8).

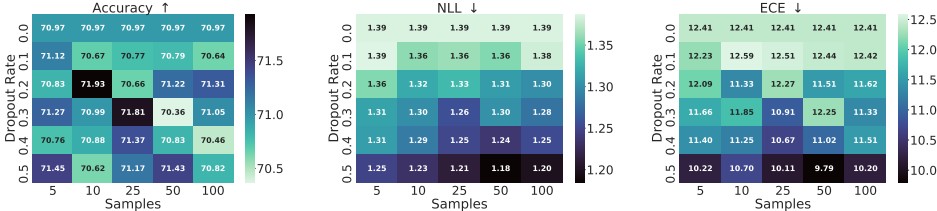

Figure 7: Performing Monte Carlo Dropout on UMBC+Set Transformer slots leads to increases in accuracy, NLL, ECE. The top row corresponds to a 0% dropout rate and is constant over dropout sample sizes. Experiment uses ModelNet40 with test set size of 100. Figures for set sizes 1000 and 2048 can be found in Figures 11 and 12

ModelNet40-C results can be seen in Figure 6. UMBC+ models give strong ECE performance in all test set sizes, improving over non-MBC baselines, especially for test set size 100 and UMBC+Set Transformer where the largest miscalibration in baseline models is.

## 6 CONCLUSION

In this work, we have shown that composing a set function consisting of a MBC base function $f$, with an arbitrary set function head $f^*$, we can make the composition $F = f^* \circ f$ universally mini-batch consistent (UMBC). We have provided proofs in Theorem 4.1, empirical experiments, and unit tests (included in the supplementary file) which prove our assertions. Likewise we have loosened the known constraints on the structure of the SSE Proposition 4.1, establishing an equivalency to the PMA layers of the Set Transformer. We have demonstrated that there are cases where a UMBC $F$ outperforms previous simpler MBC models, and explored an interesting MBC dropout strategy made possible by UMBC which leads to improved calibration and NLL. As the field of set-functions continues to widen, we look forward to seeing future research in the area of MBC set functions.

## 7 REPRODUCIBILITY STATEMENT

In addition to the details listed below, all code has been included as supplementary material and at the following URL where animations can be viewed in any broswer: `https://github.com/anonymous-subm1t/umbc`. Further descriptions of the construction of the Mixture of Gaussians dataset can be found in Appendix B, details on ImageNet clustering can be found in Appendix E, further training details regarding optimization and model settings can be found in Appendices G and H respectively.

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

# A APPENDIX

We will briefly describe the contents of each section of this appendix below:

- Appendix B: Extra information and results related to MoG Amortized Clustering.
- Appendix C: Details of the experiment depicted in Figure 1.
- Appendix D: A note on MBC testing of the Set Transformer.
- Appendix E: Details on ImageNet amortized clustering.
- Appendix F: A note on the UMBC attention softmax stability.
- Appendix G: Training parameters/setup.
- Appendix H: Model hyperparameters/setup.
- Appendix I: Additional ablation study results/discussion.
- Appendix J: Additional results/discussion for ModelNet40 experiments.
- Appendix K Extra results augmenting MBC models with UMBC.
- Appendix L Limitations and Future Work.
- Appendix M Attention Activation Effects on Calibration.

# B DETAILS ON THE MIXTURE OF GAUSSIANS AMORTIZED CLUSTERING EXPERIMENT

We used a modified version of the MoG amortized clustering dataset which was used by Lee et al. (2019). We modified the experiment, adding random variance into the procedure in order to make a more difficult dataset. Specifically, to sample a single task for a problem with $K$ classes,

1. Sample set size for the batch $N \sim U(\text{train set size}/2, \text{train set size})$.
2. Sample class priors $\pi \sim \text{Dirichlet}([1_1, ..., 1_K])$.
3. Sample class labels $z_i \sim \text{Categorical}(\pi)$ for $i = 1, ..., N$.
4. Generate cluster centers $\boldsymbol{\mu}_{i,j} \sim U(-4, 4)$ for $i = 1, ..., K$ and $j = 1, 2$.
5. Generate cluster covariances $\boldsymbol{\sigma}_{ij} = U(0.3, 0.6)$ for $i = 1, ..., K$ and $j = 1, 2$. Then make a covariance matrix $\boldsymbol{\Sigma}_i$ for each class with $\boldsymbol{\sigma}_i$ as the diagonal.
6. Sample data $\mathbf{x}_{ij} \sim \mathcal{N}(\boldsymbol{\mu}_i, \boldsymbol{\Sigma}_i)$

In our MoG experiments, we set $K = 4$.

The Motivational Example in Figure 2 also used the MoG dataset, and performed MBC testing of the set transformer corresponding to the procedure outlined in Appendix D

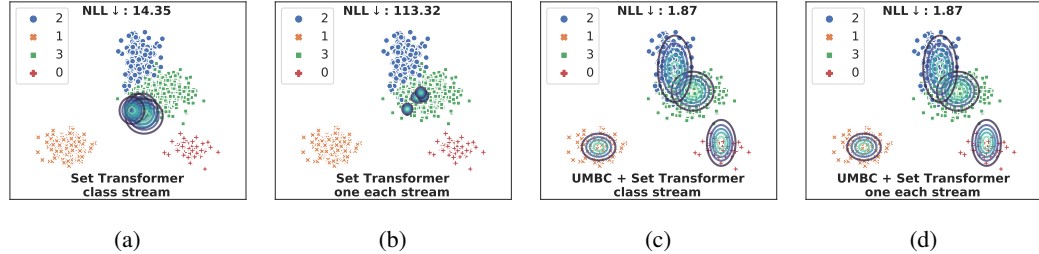

(a)      (b)      (c)      (d)

Figure 8: **a-b**: The Set Transformer struggles to perform well on different streams of a set. **c-d**: Our UMBC Module, UMBC+Set Transformer makes the Set Transformer an MBC function, yielding the same prediction regardless of the data stream. For a description of streaming settings, see Section 5, additional streams shown in Figure 2

### B.1 STREAMING SETTINGS

The four total streaming settings in Figures 2 and 8 can be described as:

- **single point stream** $\rightarrow$ streams each point in the set one by one. This causes the most severe underperformance by the Set Transformer.

- **class stream** $\rightarrow$ streams an entire class at once. The attention modules within Set Transformer cannot compare the input class with any other clusters, thereby degrading performance of Set Transformer.

- **chunk stream** $\rightarrow$ streams 8 random points at a time from the dataset, Providing limited information to the Set Transformer's attention.

- **one each stream** $\rightarrow$ streams a set consisting of a single instance from each class. Set Transformer can see examples of each class, but with a limited sample size, the encoding fails to make accurate predictions.

## C  MEASURING THE VARIANCE OF POOLED FEATURES

In Figure 1, we show the direct quantitative effect on the pooled representation when using the original Set Transformer and with our UMBC module added, UMBC+Set Transformer. The UMBC model variance is always effectively 0, while the Set Transformer gives different results for different set partition chunk sizes. The downward slope of the Set Transformer line can be explained by the fact that as the chunk size gets larger, the pooled representation will become closer to that of the full set. The procedure for MBC testing of the Set Transformer is outlined in Appendix D.

To perform this experiment, we used a randomly initialized model with 128 hidden units, and sampled a random normal input with a set size of 1024, $\mathbf{X} \in \mathbb{R}^{1 \times 1024 \times 128}$. We then created 100 random permutations of the set elements of the input and split each permutation into partitions with various chunk sizes $C_i$ where the cardinality $|C_i| \in \{2^i\}_{i=1}^6$. We then encode the whole set for each chunk size and report the observed variance between the 100 different random partitions at the various chunk sizes in Figure 1. Note that the encoded set representation is a vector and Figure 1 shows a scalar value. To achieve this, we take the feature-wise variance over the 100 encodings and report the mean over each feature. Specifically, with $\mathbf{Z} \in \mathbb{R}^{100 \times 128}$ representing all 100 encodings, $\mathbf{z} = \text{var}(\mathbf{Z})$, with $\mathbf{z} \in \mathbb{R}^{128}$. We then achieve the y values in Figure 1 by a simple mean over the feature dimension,

$$y = \frac{1}{128} \sum_i z_i \tag{6}$$

## D  A NOTE ON MBC TESTING OF THE SET TRANSFORMER

In some illustrative experiments Figures 1 and 2, we apply MBC testing to the Set Transformer to study the effects of using a non-MBC model in an MBC setting. The Set Transformer does not have a prescribed way to do this in the original work, so we took the approach of processing each chunk up until the pooled representation that results from the PMA layer. We then performed a mean pooling operation over the chunks in the following way, with $Z$ representing the final mini-batch pooled features,

$$Z = \frac{1}{N} \sum_{j=1}^P \text{PMA}(X_j) \tag{7}$$

## E  DETAILS ON THE IMAGENET AMORTIZED CLUSTERING EXPERIMENT

For the ImageNet amortized clustering experiment outlined in Section 5, we first extracted the features up until the last hidden representation and before the final linear classifier layer of the pretrained

and frozen ResNet50. These features $\mathbf{x}_i \in \mathbb{R}^{2048}$ are of a large dimension which would create excessively large linear layers for this experiment. Therefore, we projected the features down to a lower dimension $\hat{\mathbf{x}} \in \mathbb{R}^{512}$ using a random orthogonal Gaussian matrix. As this random Gaussian projection is suitable for random feature kernels (Rahimi & Recht, 2007), it should preserve the distances between points required for effective clustering with a marginal effect on overall clustering performance. To validate this assumption, we ran the Empirical model (which computes the empirical cluster mean and diagonal covariance) on both the original features $\mathbf{x}$ and the projected features $g(\hat{\mathbf{x}})$ and present the results in the table above.

To construct the ImageNet dataset, we first initialized and saved the random Gaussian projection matrix, and proceeded to process the entire ImageNet1k training set with the saved matrix. From these extracted and projected features, we chose a fixed 80/20 split for our train/test sets. Class indices for the train/text sets can be found in the supplementary file.

| Version | ARI |
|---------|-----|
| $\mathbf{z}_i \in \mathbb{R}^{2048}$ | 45.93±0.12 |
| $g(\mathbf{z}_i) \in \mathbb{R}^{512}$ | 44.09±0.11 |

## F  NUMERICAL STABILITY OF MBC SOFTMAX ATTENTION ACTIVATION

Numerical stability of the softmax requires that the values are not allowed to overflow. Generally this is done by subtracting the maximum value from all softmax logits which allows a stable and equivalent computation.

$$\frac{e^{x-\max(\mathbf{x})}}{\sum_{x' \in \mathbf{x}} e^{x'-\max(\mathbf{x})}} = \frac{e^x e^{-\max(\mathbf{x})}}{e^{-\max(\mathbf{x})} \sum_{x' \in \mathbf{x}} e^{x'}} = \frac{e^x}{\sum_{x' \in \mathbf{x}} e^{x'}} \tag{8}$$

This poses a problem when using the plain softmax attention activation, as the $\max(.)$ in Equation (8) requires a max over the whole set of $N$ items which is unknowable given the current mini-batch.

Originally, we had devised a special conditional update rule which would maintain the same form as in Equation (8), by tracking the overall max of each row of the attention matrix and then conditionally updating either the current $A$ and $\zeta$ or the previously stored values from the last processed partition. Those updates needed to be calculated in the exponential space which cause a propagation of numerical errors through the network, becoming large enough to interfere with inference. In our experiments, we found it sufficient to calculate the softmax as a simple exponential activation with a subsequent sum over $N$ with no consideration for numerical stability. If numerical stability is a concern, one could also set a hyperparameter $\lambda$ for the model such that the softmax is calculated with an exponential function such as $e^{z_i - \lambda}$, which should provide a reasonable solution.

## G  TRAINING SPECIFICATION

We use no L2 regularization, except for the ModelNet40 experiments, which use a small weight decay of $1e-7$. This was a setting taken from previous experiments by Lee et al. (2019); Zaheer et al. (2017) which used dropout before and after the pooling layers and other regularization strategies such as gradient clipping to avoid overfitting.

The only experiment which utilized any kind of data augmentations was the ModelNet40 experiments which used random rotations of the point cloud as is common in the precedent experiments (Zaheer et al., 2017; Lee et al., 2019; Bruno et al., 2021)

All single runs of all of our experiments were able to fit on a single GPU with 12GB of memory.

Table 5: The hyperparameter setup for all of our experiments involving UMBC modules.

| | Experiments | | |
|--------|------|----------|-----------|
| Setting | MoG | ImageNet | ModelNet40 |
| Optimizer | Adam | Adam | Adam |
| Learning Rate | 1e-3 | 1e-3 | 1e-3 |
| Data Augmentation | ✗ | ✗ | ✓ |
| Epochs | 50 | 50 | 1000 |
| Iters/Epoch | 1000 | 1000 | 9840 |

## H UNIVERSAL MODEL SPECIFICATION

Unless otherwise specified, all universal modules were run with the following model hyperparameter settings in Table 6. The settings for the MoG dataset apply to those in Figure 4, and Figure 5 studies the effects of changing individual settings.

Table 6: The hyperparameter setup for all of our experiments involving UMBC modules. The hyperparameters were chosen as sensible default based on previous architectures in Lee et al. (2019); Zaheer et al. (2017); Kim (2021)

| | Experiments | | |
| --- | --- | --- | --- |
| Setting | MoG | ImageNet | ModelNet40 |
| Embedder | ✓ | ✗ | ✓ |
| Hidden dim | 128 | 256 | 256 |
| Num. Slots Per Parallel UMBC | 128 | 32 | 64 |
| Slot-type | random | random | random |
| Slot LayerNorm | ✓ | ✓ | ✓ |
| FF LayerNorm | ✓ | ✓ | ✓ |
| Heads | 4 | 4 | 4 |
| Slot Dropout Prob. | 0% | 50% | 50% |
| Attention Activation | softmax | softmax | softmax |
| Slot Residual | ✓ | ✓ | ✓ |
| UMBC Num. Parallel | 1 | 4 | 4 |
| Test MC Samples | 10 | 100 | 10 |

**Slots** Different from both Locatello et al. (2020) and Bruno et al. (2021), we use unique initial slot parameters for each slot such that the set of slots $S \in \mathbb{R}^{K \times d}$ has a separate parameter for each $k_i \in K$. We do this because the original Slot Attention in (Locatello et al., 2020) used a GRU in an inner loop to adapt the single general slot into specific slots for a given task, forcing them to 'compete' to capture different parts of the input. We cannot use a GRU, as it violates Property 3.3, so we instead let each slot $k_i \in K$ learn to adapt to the overall data distribution. We always used the same dimension of inputs $X$ and slots $S$.

**Random Slots** To initialize the random Gaussian slots, we use a similar initialization strategy as (Blundell et al., 2015) and initialize $\mu \in \mathcal{U}[-0.2, 0.2]$ and $\log \sigma \in \mathcal{U}[-5.0, -4.0]$. During training, we sample the distribution with reparameterization $\mathbf{s}_k = \boldsymbol{\mu}_k + \boldsymbol{\sigma}_k * \boldsymbol{\epsilon}_k$ with $\boldsymbol{\epsilon}_k \sim \mathcal{N}(0, I^d)$.

**Embedder** We found it useful to place a single layer embedding function at the base of UMBC modules which consists of a single linear layer and a ReLU activation function. We used this embedder in all experiments except the ImageNet amortized clustering, as the ResNet feature extractor acted as the embedding function in this case.

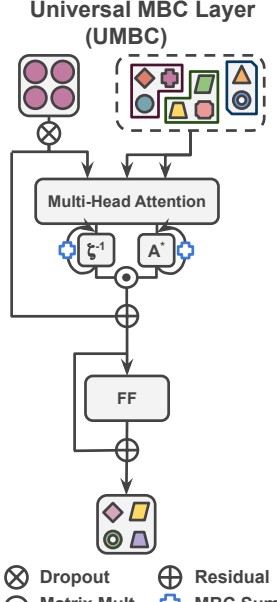

Figure 9: The architecture of a UMBC layer. $A^*$ represents the unnormalized attention matrix $\sigma(S X_i^\top) X_i$ discussed in Proposition 4.1 and 'MBC Sum' represents the summation in Equation (3)

## I ADDITIONAL ABLATION RESULTS

In addition to the results in Figure 5, we also did an experiment looking at the effect of the number of attention heads in the UMBC layer in Figure 10. This result was uninformative, but we choose to use a stock setting of 4 attention heads in our experiment as was common in the experiments performed by Lee et al. (2019).

## J ADDITIONAL MODELNET/MODELNET-C RESULTS

Table 8 shows extra results from the ModelNet point cloud classification task. In this table, we include results for 'UMBC+SSE' and 'UMBC+Deep Sets' for completeness. While there is a slight decrease in accuracy for both 'UMBC+SSE' and 'UMBC+Deep Sets,' UMBC improves SSE in terms of NLL and ECE while lowering the performance of Deep Sets. This seems to generally agree with the results in Figure 4, indicating that it is likely unhelpful to add a UMBC $f$ to an already MBC $f^*$, and instead the model $f^*$ should be chosen according to the given task first, and then UMBC considered if MBC treatment will be necessary.

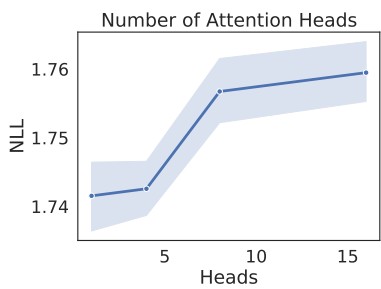

Figure 10: Ablation study on the numbers of attention heads in UMBC layers

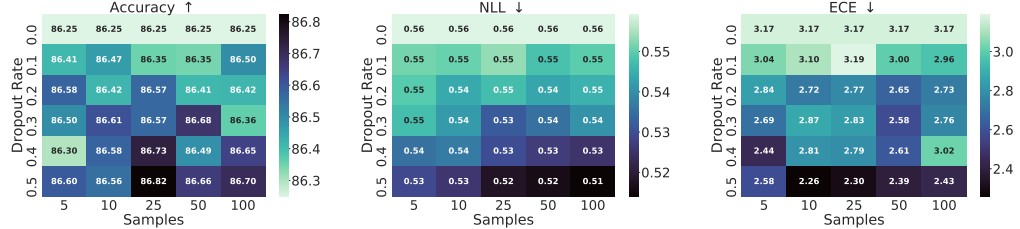

Figure 11: Performing Monte Carlo Dropout on UMBC+Set Transformer slots leads to increases in accuracy, NLL, ECE. The top row corresponds to a 0% dropout rate and is constant over dropout sample sizes. Experiment uses ModelNet40 with test set size of 1000.

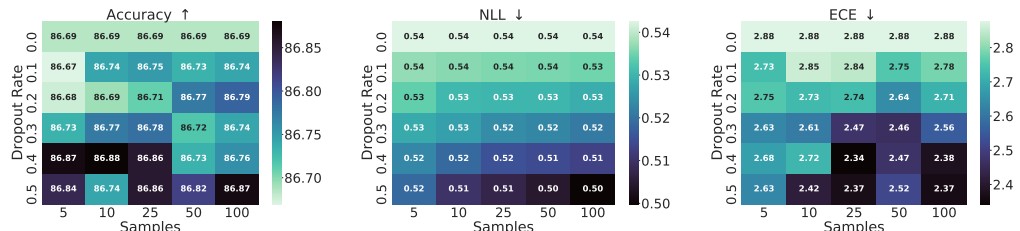

Figure 12: Performing Monte Carlo Dropout on UMBC+Set Transformer slots leads to increases in accuracy, NLL, ECE. The top row corresponds to a 0% dropout rate and is constant over dropout sample sizes. Experiment uses ModelNet40 with test set size of 2048.

ModelNet40 is prone to overfitting, and previous experiments in Deep Sets (Zaheer et al., 2017) and Set Transformer (Lee et al., 2019) have used Dropout layers both before and after the pooling function in their encoders. To evaluate the regularization effect of our dropout strategy, the last block of Table 8 includes UMBC models trained without dropout. Training without dropout generally lowers test set performance in all metric categories.

For examples of the corrupted point clouds, we refer the reader to the original work which proposed ModelNet40-C (Ren et al., 2022). In Figures 13 and 14 we provide additional boxplots for accuracy and NLL metrics which correspond to the ECE metric reported in Figure 6. In Figures 15 to 17 we provide individual boxplots for each individual corruption on accuracy, ECE, and NLL respectively. The aggregate of all of these datapoints forms the boxplots seen in Figures 6, 13 and 14. Size is reduced to avoid excessive page length. Best viewed on screen with a high zoom.

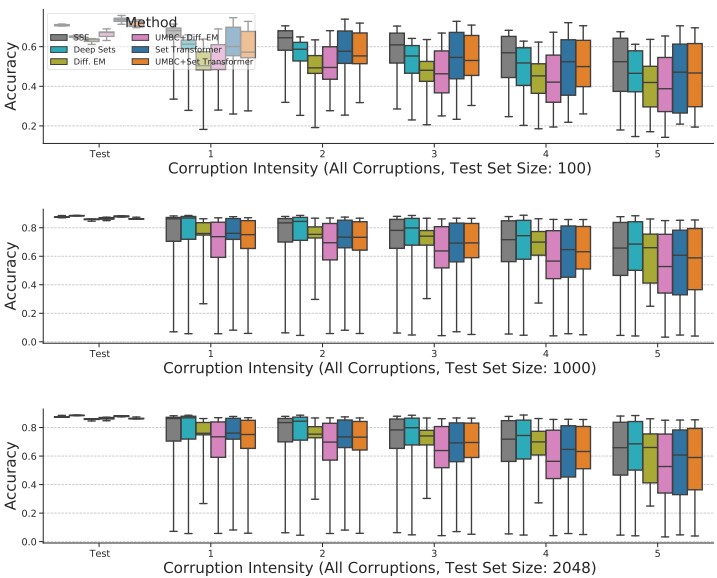

Figure 13: Accuracy across all corruptions in the ModelNet40-C dataset. This figure corresponds to the ECE results presented in Figures 6 and 14

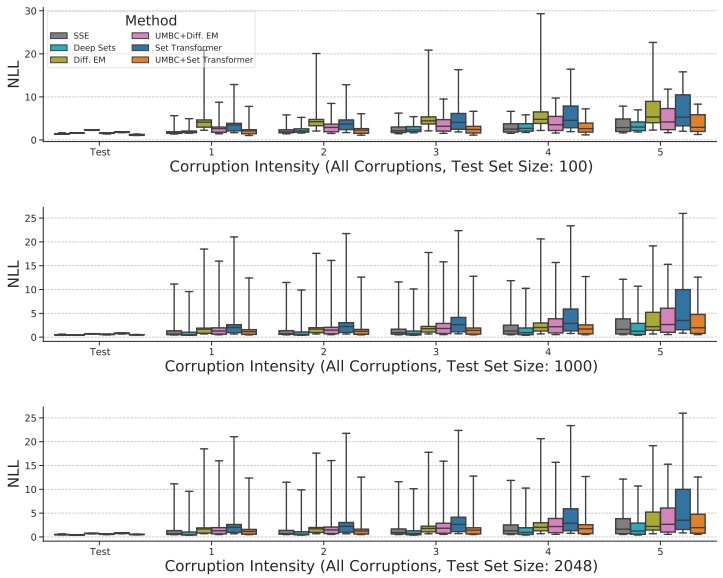

Figure 14: NLL across all corruptions in the ModelNet40-C dataset. This figure corresponds to the ECE results presented in Figures 6 and 13

# K ADDING THE UMBC MODULE TO EXISTING MBC FUNCTIONS

# L LIMITATIONS & FUTURE WORK

**UMBC is a bottleneck** UMBC projects the input set to a fixed size, and can therefore be a bottleneck, causing possible loss of information from the input set. An interesting line of research could be an exploration of methods to maximize mutual information between the input set of cardinality $N$ and the projected set of cardinality $K$, or an exploration of other forms which a UMBC may take, we look forward to seeing future research in this area.

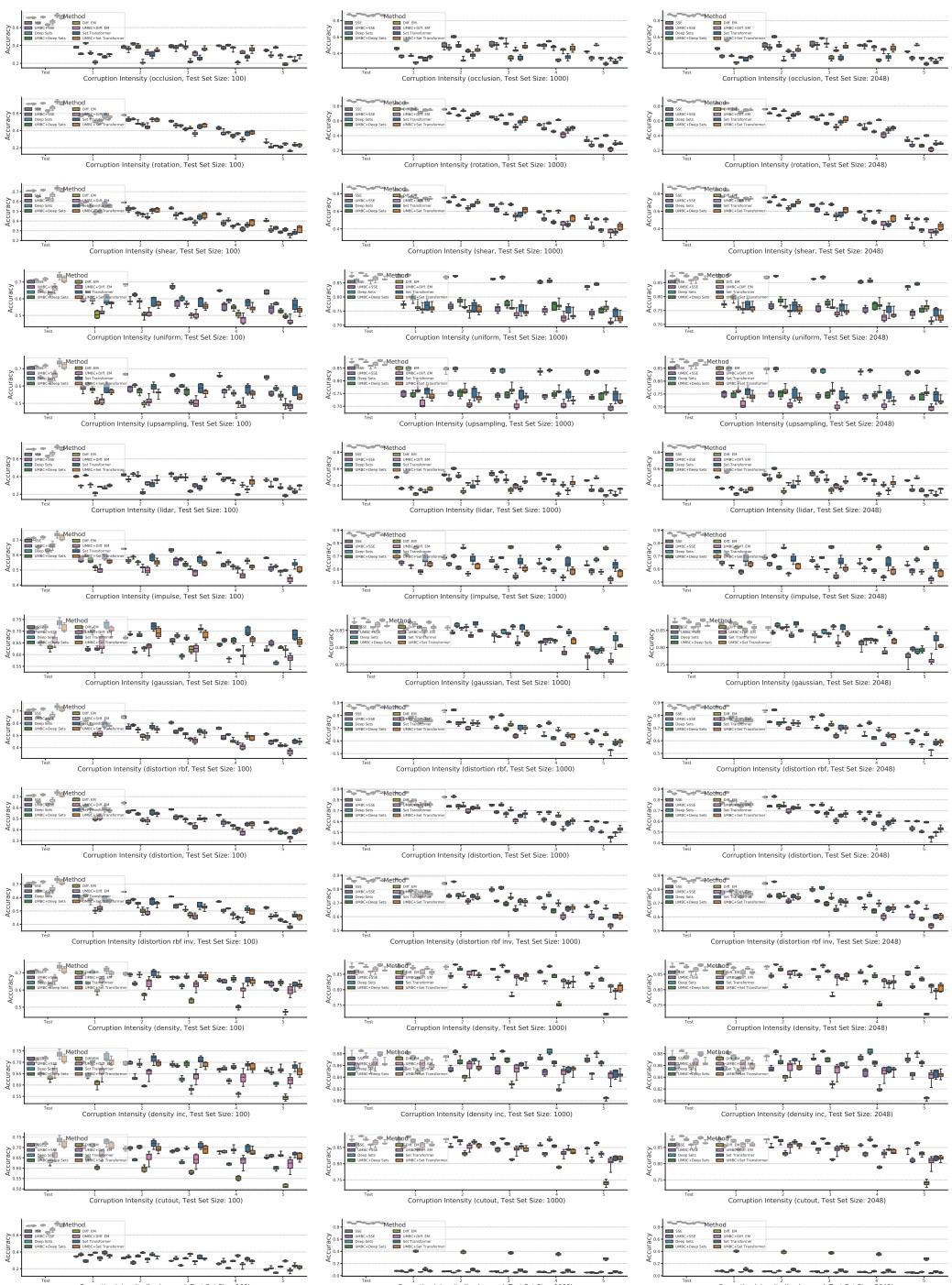

Figure 15: Accuracy boxplots for individual ModelNet-C test results. Size is minimized to avoid excessive page length. Best viewed on screen with high zoom

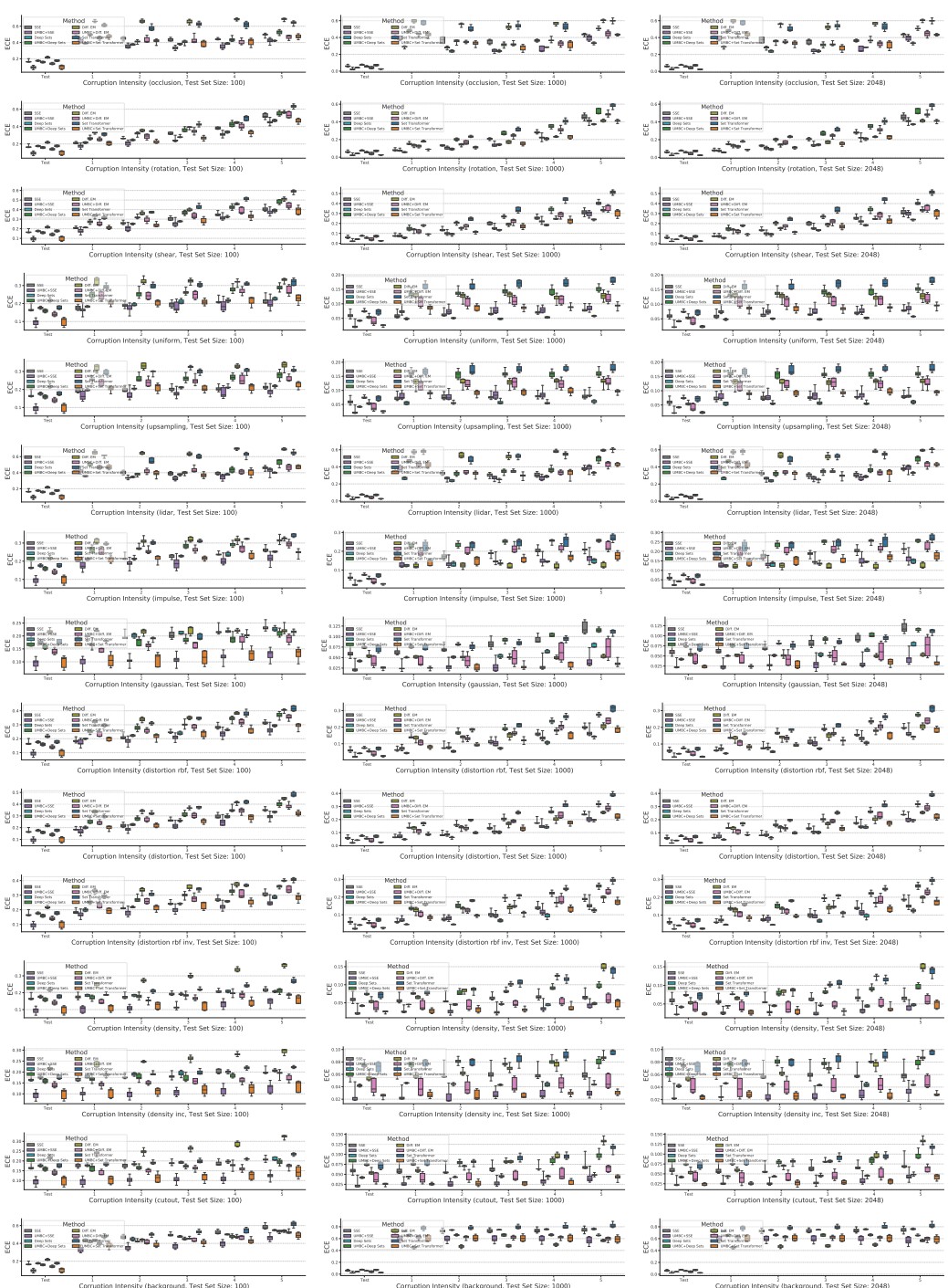

Figure 16: ECE boxplots for individual ModelNet-C test results. Size is minimized to avoid excessive page length. Best viewed on screen with high zoom

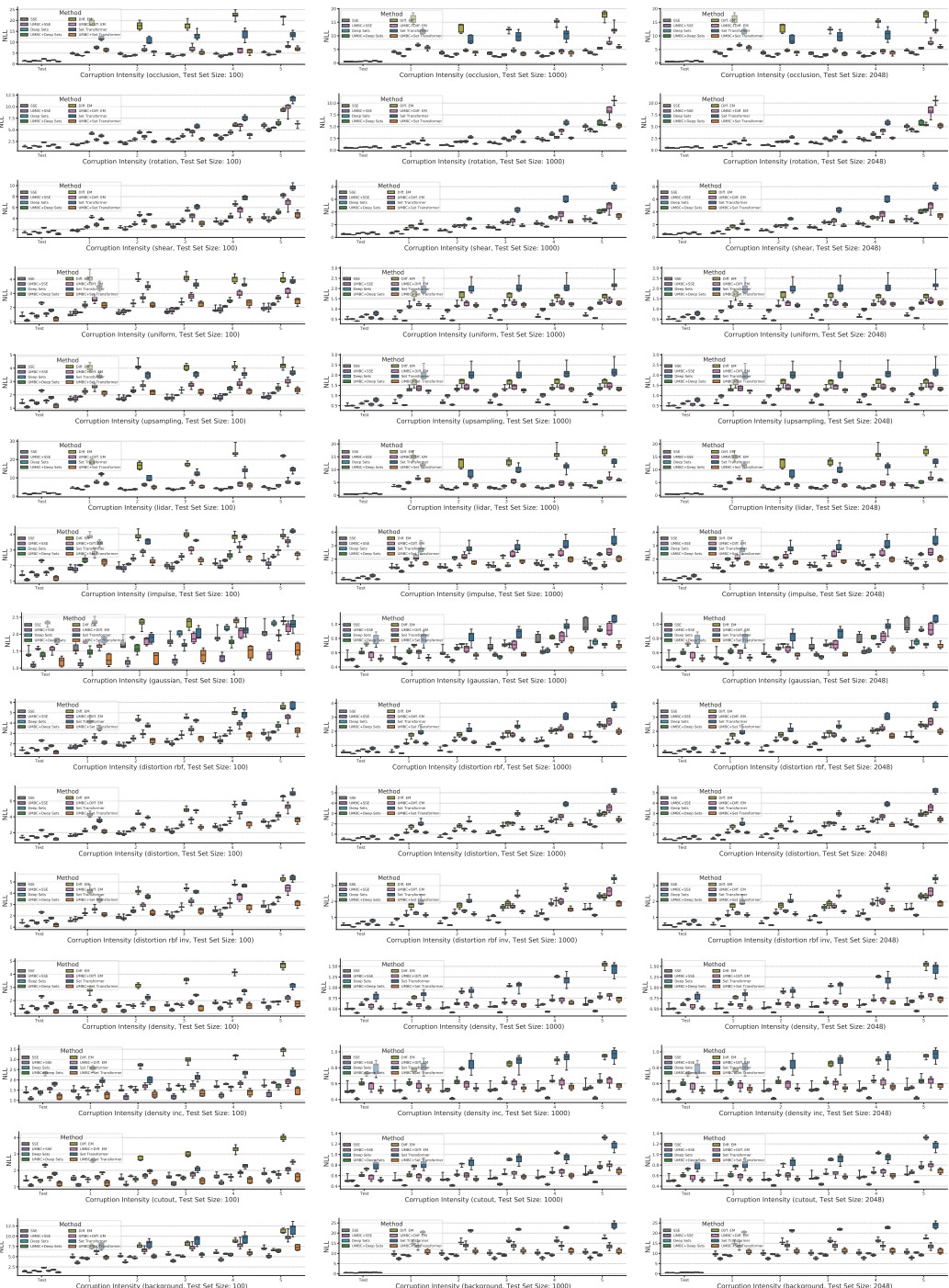

Figure 17: NLL boxplots for individual ModelNet-C test results. Size is minimized to avoid excessive page length. Best viewed on screen with high zoom

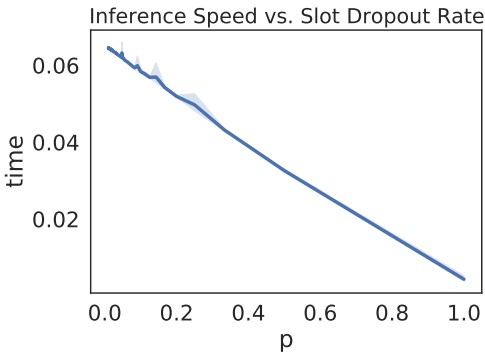

Figure 18: Because of the model structure, higher slot dropout rates correspond to faster training times, and smaller set sizes as input to the subsequent set encoder modules. A dropout rate of $p = 0.5$ in the function $f$ will, in expectation, deliver a set size of $K/2$ to the subsequent function $f^*$. This figure was generated from a UMBC+Set Transformer model with 128 hidden units and input $\mathbf{x} \in \mathbb{R}^{32 \times 200 \times d}$. The plotted line shows mean and standard deviation for 250 iterations at each $p \in [1, 99]$. As a safeguard against unstable training, we ensure that at least one slot remains after dropout is applied.

Table 7: Amortized Clustering on ImageNet features extracted with a pre-trained ResNet50.

| Model | NLL ↓ | ARI ↑ |
|---|---|---|
| Empirical | 1028.22±1.24 | 44.09±0.11 |
| Deep Sets | 531.44±0.15 | 6.18±0.08 |
| SSE | 520.29±0.63 | 22.91±1.85 |
| Set Transformer | 512.59±0.33 | 17.13±3.67 |
| UMBC+Deep Sets | 532.87±0.69 | 6.22±0.18 |
| UMBC+SSE | 544.67±3.64 | 16.59±1.26 |
| UMBC+Set Transformer | **503.89±0.87** | **23.68±1.85** |

**Train/Test Set Size Variability** In Figure 4, Deep Sets shows the tightest grouping between training set sizes, although giving the lowest overall performance, indicating that more complicated set functions which make pairwise comparisons may be less robust to varying training set sizes, which may provide an interesting topic of future research.

**Bayesian Slots** In our experiments, we used a similar random slot parameter initialization as Blundell et al. (2015). Following Bruno et al. (2021), we use no Bayesian prior on these random slots, so the increased performance of random slots is likely due to randomness aiding in exploration of the parameter space rather than learning a proper Bayesian posterior. Future work could explore the effects of incorporating a prior distribution over slots or slot dropout rates (*e.g.* Concrete Dropout (Gal et al., 2017)). This could lead to further increases in robustness to corruptions and varying set sizes.

**Large Train-Time Set Sizes** The setting we have considered is one where the train-time set size is known and the test-time setting presents severely constrained computataional resources or extremely large set sizes which require processing the set in chunks. An open problem, however, is how to handle larger-than-memory set sizes at train time, as backpropagation requires memory allocation for each element in the sets. We look forward to seeing future research which may tackle this problem, and allow for training with extremely large set sizes.

## M   ATTENTION ACTIVATIONS & CALIBRATION

To test the effect of training with different attention activation functions on calibration, we train and evaluate the UMBC+Set Transformer model on all corruptions of ModelNet40-C in Figures 19 to 21, and individual corruptions in Figures 22 to 24. Besides the change in attention activation, each

Table 8: Point cloud classification on ModelNet40. All models are trained on a set size of 1000 randomly sampled points, and evaluated on 100, 1000, and 2048 (max) test set sizes. UMBC models in the second block are trained and tested with our slot dropout technique outlined in Section 4. Models in the first block are trained without Slot Dropout, and use all available slots output from $f$ at both train and test time

| Model | Accuracy ↑ | | | NLL ↓ | | | ECE ↓ | | |
| --- | --- | --- | --- | --- | --- | --- | --- | --- | --- |
| | 100 | 1000 | 2048 | 100 | 1000 | 2048 | 100 | 1000 | 2048 |
| UMBC+Deep Sets (No Dropout train) | 69.96±0.64 | 87.50±0.21 | 87.58±0.16 | 1.82±0.06 | 0.66±0.02 | 0.64±0.02 | 21.25±0.54 | 8.59±0.32 | 8.51±0.26 |
| UMBC+SSE (No Dropout train) | 68.80±1.00 | 84.81±1.17 | 84.89±1.39 | 1.19±0.06 | 0.55±0.04 | 0.54±0.04 | 11.80±2.07 | 3.05±0.64 | 3.02±0.82 |
| UMBC+Set Transformer (No Dropout train) | 71.52±0.75 | 86.56±0.47 | 86.61±0.45 | 1.50±0.43 | 0.63±0.14 | 0.62±0.15 | 13.36±4.66 | 4.22±2.04 | 4.28±2.09 |
| UMBC+Deep Sets | 71.53±1.03 | 87.52±0.25 | 87.74±0.45 | 1.48±0.09 | 0.61±0.03 | 0.62±0.03 | 16.39±1.52 | 7.53±0.38 | 7.49±0.50 |
| UMBC+SSE | 71.03±0.73 | 86.19±0.62 | 86.36±0.46 | 1.11±0.09 | 0.50±0.01 | 0.49±0.01 | 9.67±2.03 | 2.42±0.77 | 2.37±1.10 |
| UMBC+Set Transformer | 71.18±1.52 | 86.56±0.49 | 86.77±0.29 | 1.23±0.15 | 0.53±0.03 | 0.51±0.03 | 10.37±2.24 | 2.60±0.19 | 2.35±0.24 |

model was trained with the same settings as the UMBC+Set Transformer from the corresponding experiments in Figures 6, 13 and 14. Surprisingly, we find the slot-softmax, originally used by Locatello et al. (2020) delivers strong performance in terms of NLL and ECE, although it gives slightly lower accuracy on the natural, uncorrupted test set.

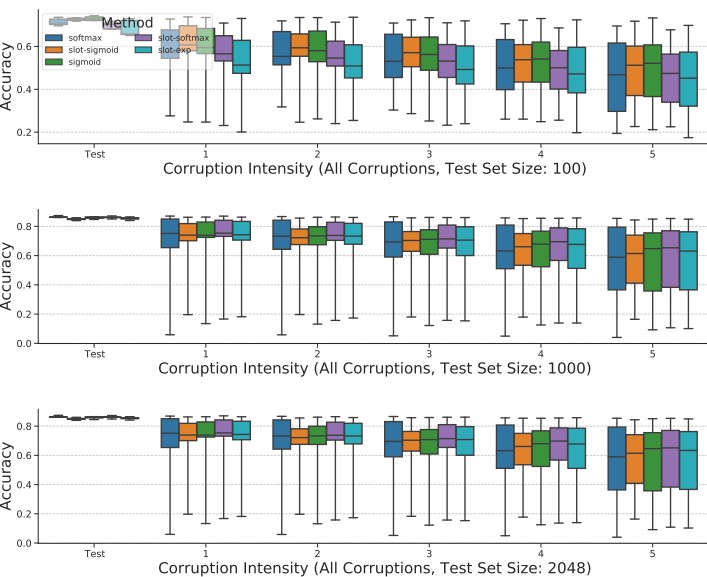

Figure 19: Accuracy across all corruptions on the ModelNet40-C dataset for UMBC+Set Transformer with different attention activation functions. This figure corresponds to the results presented in Figures 20 and 21

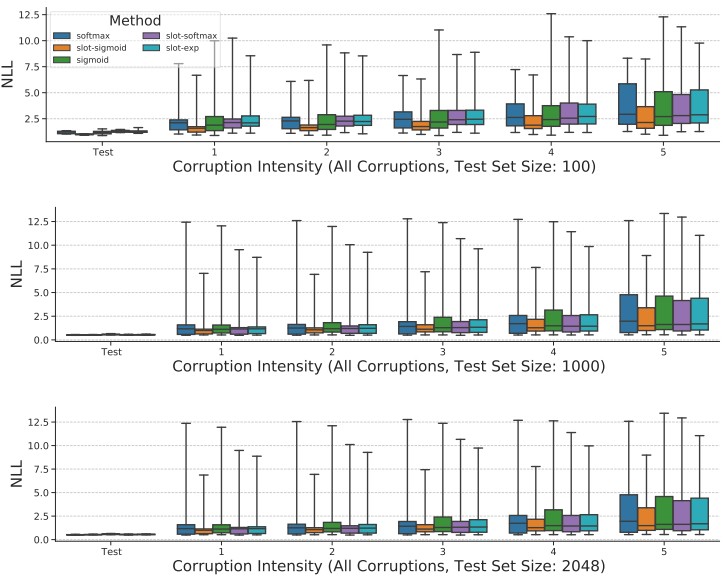

Figure 20: NLL across all corruptions on the ModelNet40-C dataset for UMBC+Set Transformer with different attention activation functions. This figure corresponds to the results presented in Figures 19 and 21

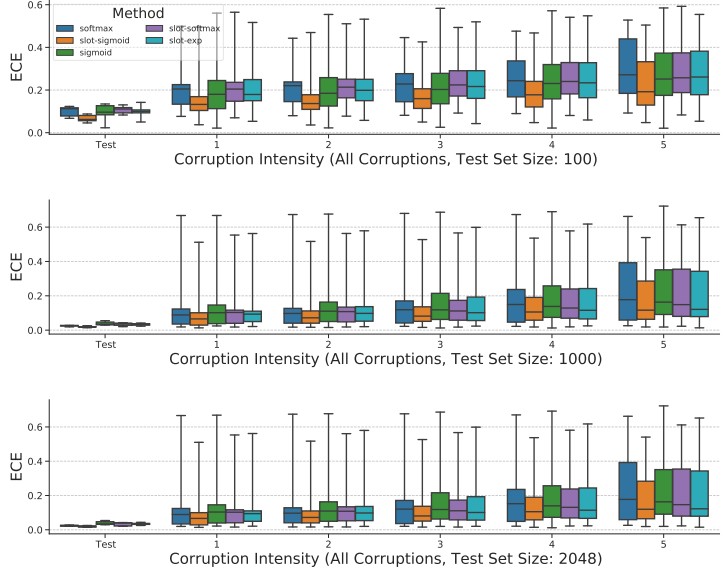

Figure 21: ECE across all corruptions on the ModelNet40-C dataset for UMBC+Set Transformer with different attention activation functions. This figure corresponds to the results presented in Figures 19 and 20

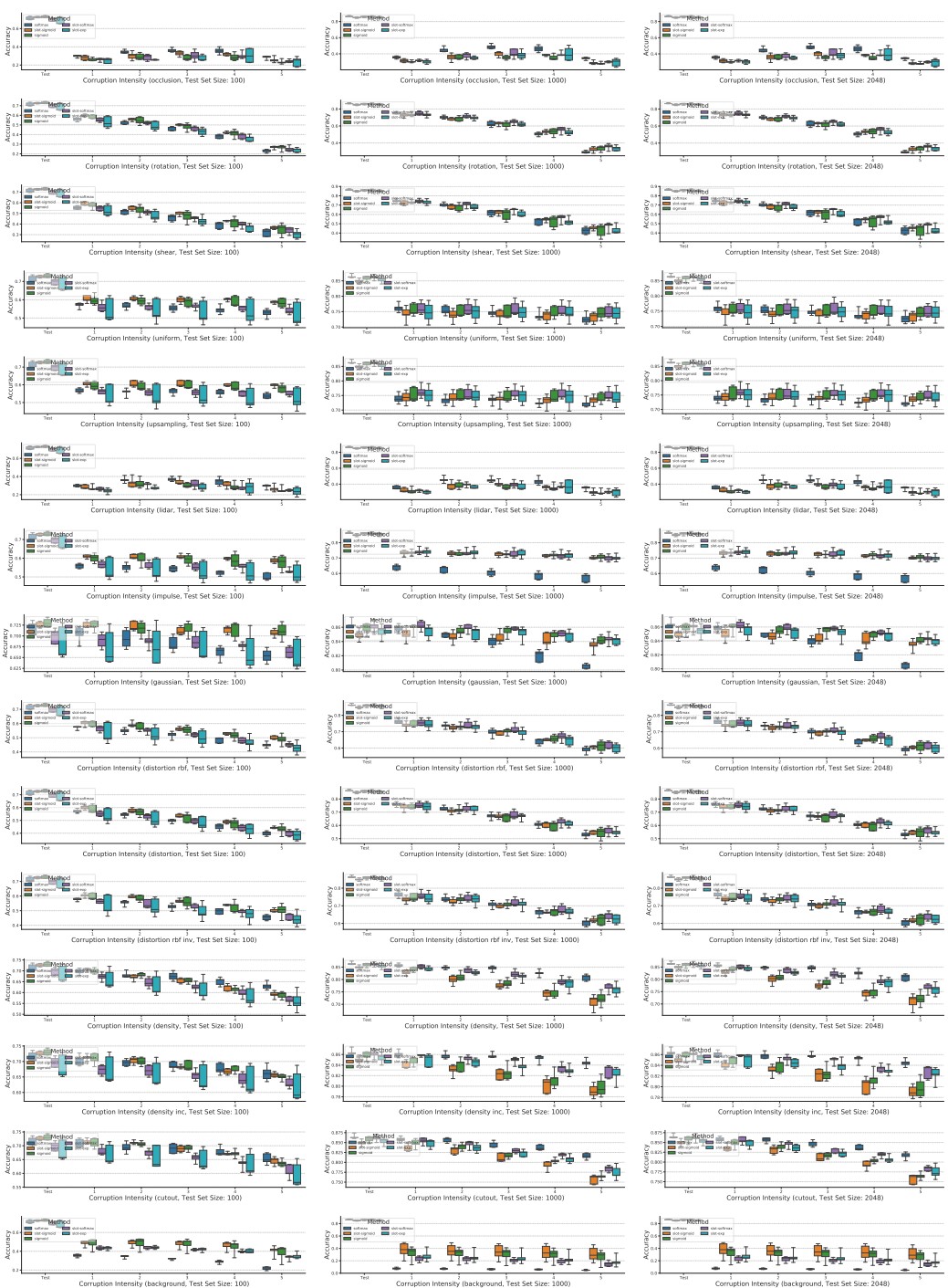

Figure 22: Accuracy boxplots for individual ModelNet-C tests with UMBC+Set Transformer and different attention activation functions. Size is minimized to avoid excessive page length. Best viewed on screen with high zoom

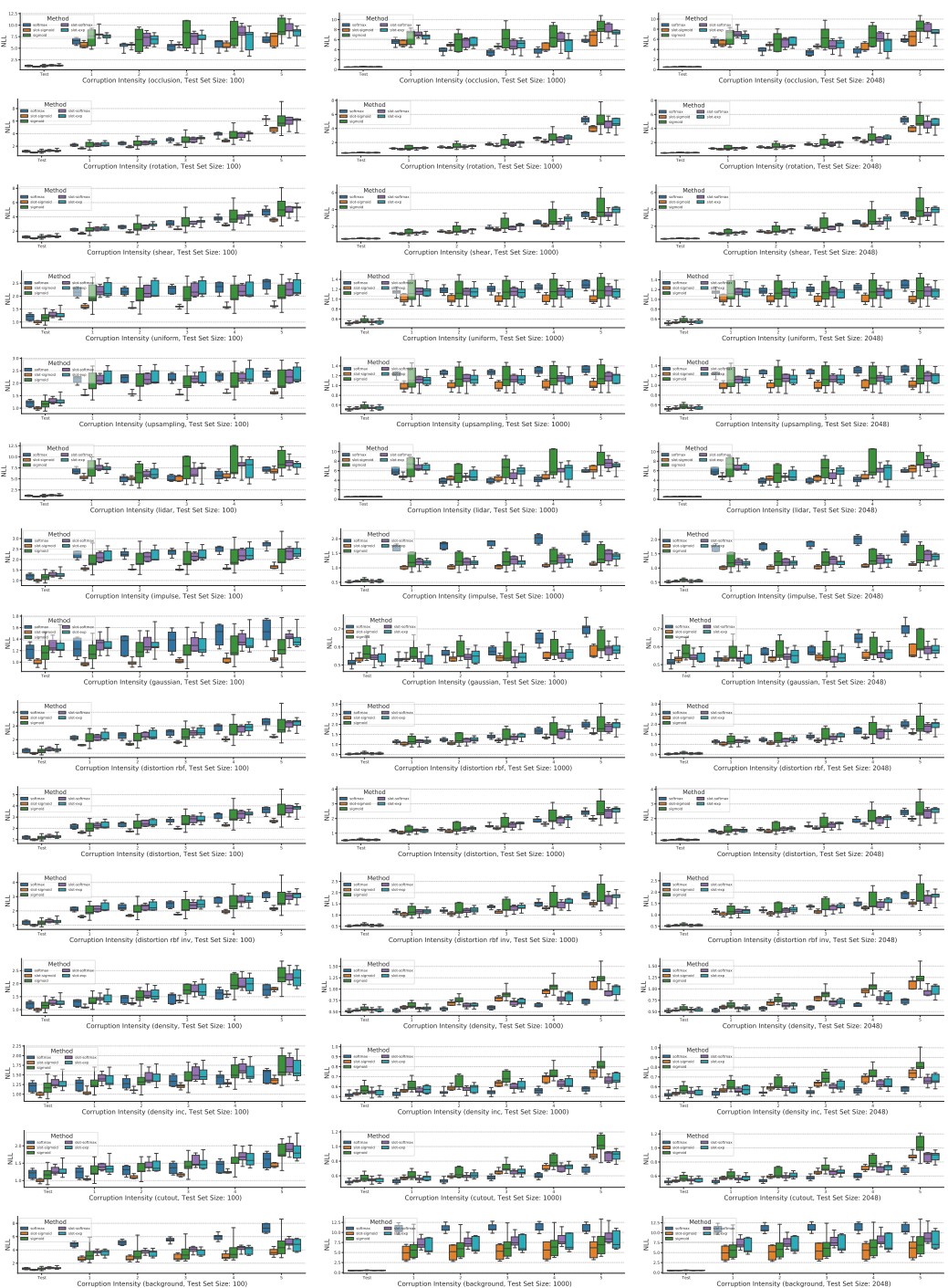

Figure 23: NLL boxplots for individual ModelNet-C tests with UMBC+Set Transformer and different attention activation functions. Size is minimized to avoid excessive page length. Best viewed on screen with high zoom

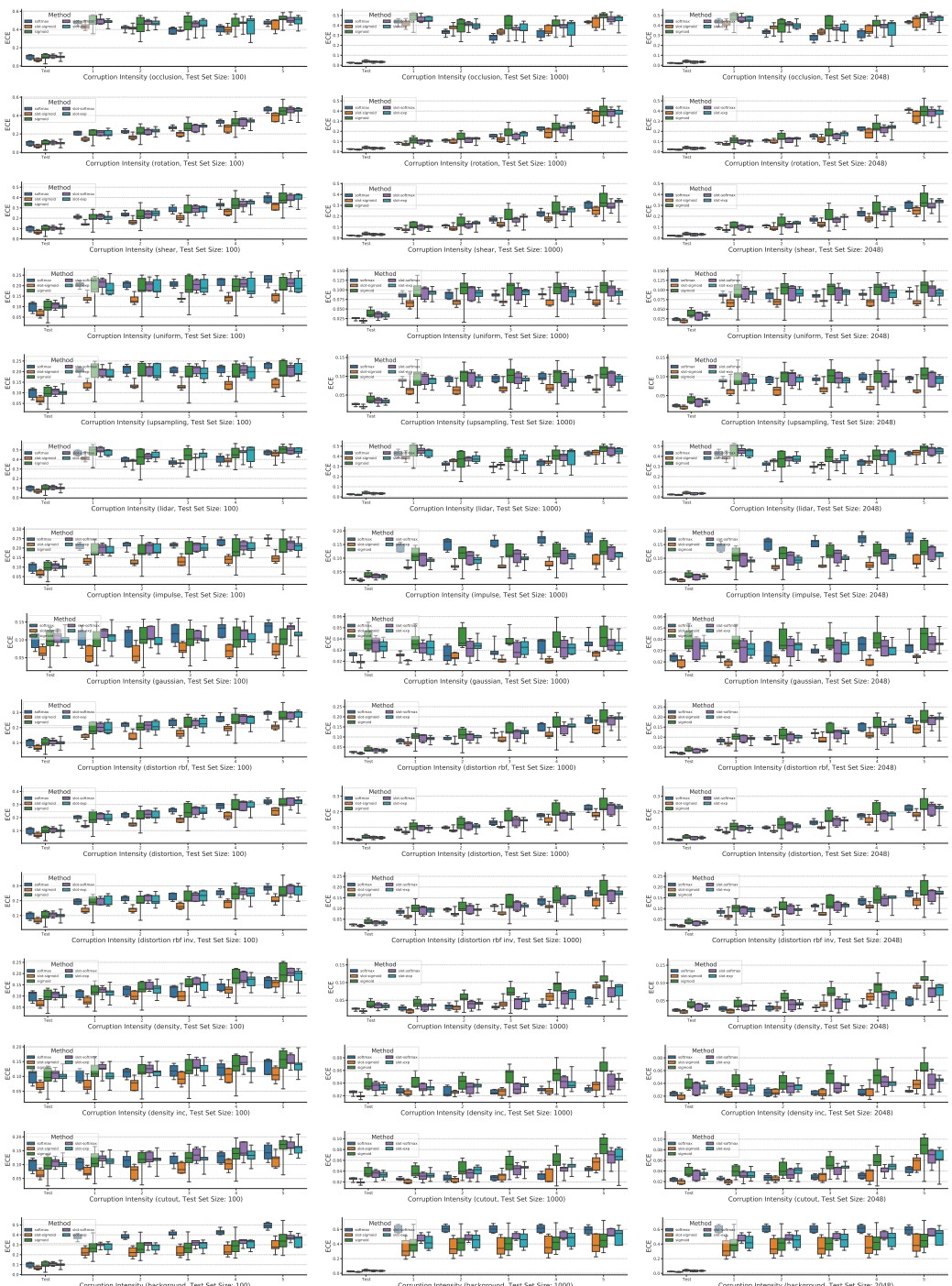

Figure 24: ECE boxplots for individual ModelNet-C tests with UMBC+Set Transformer and different attention activation functions. Size is minimized to avoid excessive page length. Best viewed on screen with high zoom

