# OpenReview forum: "Universal Mini-Batch Consistency for Set Encoding Functions"
_ICLR.cc/2023/Conference — Submitted to ICLR 2023_

### Official Review · Reviewer_SNvt · 2022-10-26

**Confidence:** 4
**Correctness:** 3
**Technical Novelty And Significance:** 2
**Empirical Novelty And Significance:** 3
**Recommendation:** 5

**Clarity, Quality, Novelty And Reproducibility:**


The writing has some space to be improved. I list some comments bellow:

- In property 3.3, the notation $f(X)=Z$ is redundant. I think you may want to define the aggregation function as $ g: \{Z_j \in \mathbb{R}^{d^\prime}\}_{j=1}^P \rightarrow \mathbb{R}^{d^\prime}$  instead of $g: \{f(X_j) \in \mathbb{R}^{d^\prime}\}_{j=1}^P \rightarrow \mathbb{R}^{d^\prime}$, or you could discard the notation $f(X)=Z$ avoid confusion.

- In the line following property 3.3, the definition of $\sigma(\cdot)$ is missing. I think it should be a sigmoid function with some normalization constraints. Moreover, it would be helpful to understand if you could explicitly write out what $g$ and $f$ stand for in $\mathrm{Attention}(S,X) = \sigma(SX^T) = \sum_{j=1}^P \sigma(SX_j^T) X_j$.

- Lemma 4.1 & Theorem 4.1 seems to be a bit redundant. It’s better to merge them together. Moreover, it would be great if there exists an algorithm to show how UMBC works. The equation $X\in \mathbb{R}^{N\times d} \rightarrow f(X) \rightarrow \Phi \in \mathbb{R}^{K \times d} \rightarrow f^* (\Phi) \rightarrow Y$  is helpful for readers to understand. However, it is confusing for me as the MBC aggregation function $g$ is missing in this equation. I think an algorithm (like alg.1 in the SSE paper) would be a better and more direct way to show how UMBC works.

- In lemma 4.1, what does an arbitrary set function stand for? If I understand correctly, it should be “arbitrary S2V function”, as a set function is the one that takes a set as input, and the S2V function is a kind of set function which satisfies permutation invariance/equivariance according to definition 3.1.

- I think a better expression in theorem 4.1 could be: “Let $g$ and $f$ be mini-batch consistent, $f^*$ be …. By Lemma 4.1, the composition ….“  Otherwise, it is quite confusing to say $f$ satisfies property 3,3 and at the same time say $g$ is mini-batch consistent.


**Strength And Weaknesses:**

### Strengths
- The problem raised in this paper is interesting. The attached code is helpful and reproducible.
- The proposed MC dropout method is nice. Despite being straightforward, it is exactly the first time to consider uncertainty estimation and model calibration in the set function communities.
### Weaknesses
- The UMBC method proposed in this paper seems to be an extended version of SSE, which might limit the technical novelty.
- It seems that the proposed UMBC architecture just composes the SSE ($g \circ f$) and a permutation invariant/equivariant set function ($f^*$) on the top. It is unclear how the additional set function (i.e., $f^*$) contributes to the expressiveness of mini-batch consistency modeling. I am afraid that if we stack multiple layers of SSE [Bruno et al. 2021], we could achieve the same goal of using one layer of SSE and a set transformer on the top.

- Proposition 4.1 seems to be an important contribution, as it releases the sigmoid constraint in SSE and shows that the softmax attention also satisfies mini-batch consistency. However, it is unclear about the contribution of softmax attention compared to the sigmoid attention. Actually, we can apply arbitrary kernel functions to calculate the attention weights between the inducing points $S$ and the data $X$. One could just replace the sigmoid function with an exponential kernel.


Bruno, Andreis, et al. "Mini-Batch Consistent Slot Set Encoder for Scalable Set Encoding." *Advances in Neural Information Processing Systems* 34 (2021): 21365-21374.


**Summary Of The Paper:**

This paper presents a new architecture to satisfy the mini-batch consistency (MBC) property, which is an important property required by set functions in the streaming fashion. Specifically, given a mini-batch consistent function $f$ (e.g., slot set encoder, SSE), the authors prove that for an arbitrary set function $f^*$, the composition $F = f^* \circ f$ is also mini-batch consistency, resulting in the proposal architecture, universal MBC set function (UMBC). Additionally, a Monte Carlo dropout strategy is proposed to cast the UMBC module more robust on out-of-distribution data. Experimental results show promising results on amortized clustering and point cloud classification.

**Summary Of The Review:**

Overall, although the problem is interesting and the method works well in some cases, I think the work needs to be improved. The major concern is the unclear contribution of the additional permutation invariant/equivariant set function $f^*$ on the top of SSE architecture. In this regard, I think the proposed method is somewhat incremental, and most claimed contributions are similar  as SSE.

---

> ### Author Response · Authors · 2022-11-14
> **Author Response (2/2)**
>
> > Proposition 4.1 seems to be an important contribution, as it releases the sigmoid constraint in SSE and shows that the softmax attention also satisfies mini-batch consistency. However, it is unclear about the contribution of softmax attention compared to the sigmoid attention. Actually, we can apply arbitrary kernel functions to calculate the attention weights between the inducing points  and the data . One could just replace the sigmoid function with an exponential kernel.
>
> - Thank you for noting the importance of the contribution. You are correct, that an exponential kernel could be used as a similarity function. In the paragraph titled "SSE's connection to PMA's," we lay out the requirements for the attention activation functions which are:
>   1. It maps the values in the attention matrix to strictly positive values, and
>   2. It has an *optional* normalization constant which can be factored over N.
> - We have added a footnote to Proposition 4.1 to indicate that any valid attention kernel function can be used.
> - We highlighted 5 possible functions in Table 2 which are straightforward combinations of existing attention activations, but a near infinite combination of functions can be used, including all kernel functions which fit the requirements.
> - **We performed an ablation study in Fig. 5 which showed softmax attention to perform better than other attention functions in Table 2.**
>
>
> ---
>
> > In property 3.3, the notation  is redundant. I think you may want to define the aggregation function as $g: \\{ Z_j \in \mathbb{R}^{d^\prime} \\}_{j=1}^P \rightarrow \mathbb{R}^{d^\prime}$ instead of
>
> > $g: \\{ f(X_j) \in \mathbb{R}^{d^\prime} \\}_{j=1}^P \rightarrow \mathbb{R}^{d^\prime}$ or you could discard the notation $f(X) = Z$ to avoid confusion.
>
> Thank you for pointing this out. We agree and we have revised the text to remove $f(X) = Z$ from Property 3.3.
>
> ---
>
> > In the line following property 3.3, the definition of $\sigma$ is missing. I think it should be a sigmoid function with some normalization constraints. Moreover, it would be helpful to understand if you could explicitly write out what $g$ and $f$ stand for in $\text{Attention}(S, X) = \sigma(SX^\top) = \sum_{j=1}^P \sigma(SX_j^\top)X_j$
>
> - In the original text, $\sigma$ was defined as a normalized sigmoid.
> - We have updated the text following Property 3.3 to include a definition of the slot normalization process and denoted $\hat{\sigma}$ as the function which applies the sigmoid AND the normalization process.
> - The revised text near Equation 1 defines $f$ as the attention operation and $g$ as the sum over partitions on the RHS of Equation 1
> - This is also included in the algorithm of the updated text.
>
> ---
>
> >Lemma 4.1 & Theorem 4.1 seems to be a bit redundant. It’s better to merge them together.
>
> We have merged them both into Theorem 4.1. Thank you.
>
> ---
>
> >It would be great if there exists an algorithm to show how UMBC works. The equation $X \in \mathbb{R}^{N \times d} \rightarrow f(X) \rightarrow \Phi \in \mathbb{R}^{K \times d} \rightarrow f^*(\Phi) \rightarrow Y$  is helpful for readers to understand. However, it is confusing for me as the MBC aggregation function  is missing in this equation. I think an algorithm (like alg. 1 in the SSE paper) would be a better and more direct way to show how UMBC works.
>
> In the revised version, we have added an algorithm to Section 4 in place of the previous equation.
>
> ---
>
> > In lemma 4.1, what does an arbitrary set function stand for? If I understand correctly, it should be “arbitrary S2V function”, as a set function is the one that takes a set as input, and the S2V function is a kind of set function which satisfies permutation invariance/equivariance according to definition 3.1.
>
> - We first must correct the statement that a "S2V function satisfies permutation invariance/equivariance according to definition 3.1"
>   - This is not true by the definitions of both properties. Prop. 3.1 states that the function is fixed for all permutations of the inputs.
>   - Property 3.2 states that any permutation applied to the inputs, results in the same permutation of the outputs.
>   - **Therefore any function which satisfies Prop. 3.1 will fail to satisfy Prop. 3.2 by definition and vice versa.**
> - We meant *arbitrary set functions* to imply that $f$ and $f^*$ satisfy either Property 3.1 (permutation invariance) or Property 3.2 (permutation equivariance).
> - We see why this might be confusing, and we have updated Theorem 4.1 (previously Lemma 4.1) to state the following:
>
> *Let $f^\ast$ be a set function satisfying either Property 3.1 or Property 3.2, and $(g, f)$ be functions satisfying Property 3.3, which together form the functional composition $F = f^\ast \circ g \circ f$. In order for $F$ to satisfy Property 3.3, it is sufficient to require the representation $Z = g\big(f(X_1), \ldots, f(X_p)\big)$ as input to $f^\ast$ to satisfy Property 3.3.*
>
> ---
>
> ### Thank you for your review, we remain open for further discussion.

---

> ### Author Response · Authors · 2022-11-14
> **Author Response (1/2)**
>
> ### Thank you for taking the time to review our work, we will address each of your points in turn below.
>
> ---
>
> > The UMBC method proposed in this paper seems to be an extended version of SSE, which might limit the technical novelty.
>
> > It seems that the proposed UMBC architecture just composes the SSE and a permutation invariant/equivariant set function $f^*$ on the top. It is unclear how the additional set function (i.e., $f^*$) contributes to the expressiveness of mini-batch consistency modeling. I am afraid that if we stack multiple layers of SSE [Bruno et al. 2021], we could achieve the same goal of using one layer of SSE and a set transformer on the top.
>
> - The main novelty of our method lies in the application of Theorem 4.1, allowing for all set-functions to be considered MBC.
> - It is not correct to say that stacking SSE's is the same as UMBC+Set Transformer. SSE's are MBC functions, which does not consider the possibility of composing MBC+non-MBC functions.
> - Additionally, SSE's only contain cross attention, we consider completely unconstrained set functions for $f^\ast$, as Theorem 4.1 proves $f^\ast$ can be unconstrained.
> - In fact [Bruno et al. 2021] included experiments evaluating a Hierarchical SSE (Fig 4(c) in [Bruno et al. 2021]) which showed no performance gain. In fact, performance became worse with a deeper hierarchy.
> - Contrast this with Fig. 4 where composing multiple functions with self attention creates a stronger model.
>
> ---
>
> > The major concern is the unclear contribution of the additional permutation invariant/equivariant set function on the top of SSE architecture. In this regard, I think the proposed method is somewhat incremental, and most claimed contributions are similar as SSE.
>
> - The additional benefit of $f^\ast$ can be seen in Fig. 4(e and f) which show that using UMBC+Set Transformer and UMBC+Diff. EM. in this setting create the best MBC models.
> - Note that the performance is increases over SSE, which results from the given task (clustering) which requires understanding complicated relationships between set elements.
> - It is important to note that this is the same task as Fig. 2 where Set Transformer showed NLL $\approx 140$ given a tough MBC setting. **Therefore we must ask, if one was given the test-time constraint from Fig. 2, which MBC model should you choose?** (UMBC+Set Transformer is the best MBC model).
> - The novelty of our method lies in the idea that set functions can be composed in such a way that a S2V function's output can be re-interpreted as a set by downstream functions which can therefore have fewer constraints on them.
> - Thus, more powerful functions can be used downstream, which results in better performance while maintaining Property 3.3.

---

### Official Review · Reviewer_9w5z · 2022-10-27

**Confidence:** 2
**Correctness:** 3
**Technical Novelty And Significance:** 2
**Empirical Novelty And Significance:** 3
**Recommendation:** 5

**Clarity, Quality, Novelty And Reproducibility:**

The paper is mostly clear. The authors also provide the code for reproducibility.

**Strength And Weaknesses:**

Strengths:
- The paper is mostly well-written, though there are some parts I feel are confusing (detailed later).

- The proposed framework for converting any non-MBC functions to MBC functions is simple and general.

Weakness:
- I am a bit confused with Lemma 4.1 and Theorem 4.1. As in Proposition 3.3, the Mini-Batch-Consistency (MBC) is defined for a set function f and an aggregation function g, instead of just a single function. So, what do you mean by F satisfy property 3.3? What's the corresponding aggregation function for F?

- Monte Carlo Slot Dropout: why does the composition of the functions $f^\star$ and the dropout mask still a set function? Once the dropout mask is given, shouldn't the permutation of the slots affect the output?


- Figure 4: It seems that UMBC + Diff. EM performs worse than Diff. EM alone, and this is also the case for UMBC + set transformer vs. Set transformer? I.e., UMBC + Diff. EM gives a larger negative log-likelihood, hence a lower log-likelihood.

- Table 1: Why does Oracle perform the worst in terms of NLL?

**Summary Of The Paper:**

This paper proposes a framework for converting any non-Mini-Batch-Consistent (non-MBC) models to an MBC model. Specifically, for any non-MBC set function, $f^\star$, we can convert it to an MBC function by plugging in an MBC function before it. This framework also enables incorporating uncertainty estimation methods, such as MC-Dropout for neural set functions. The authors also conducted extensive experiments and ablation studies for demonstrating the effectiveness of their algorithms.


**Summary Of The Review:**

Overall, I think the idea in this paper is interesting and novel, though it's not quite hard to derive. However, I have some doubts about the Lemmas and theorems in the paper as well as the empirical results, as detailed in the Strength&Weakness section. Also, I don't work in this area, so I am not able to assess the potential impact of this work.
I will consider to increase the rating if the authors can address my concerns.

---

> ### Author Response · Authors · 2022-11-14
> **Author Response**
>
> ### Thank you for taking the time to review our work, we will address each of your points in turn below.
>
> ---
>
> > I am a bit confused with Lemma 4.1 and Theorem 4.1. As in Proposition 3.3, the Mini-Batch-Consistency (MBC) is defined for a set function f and an aggregation function g, instead of just a single function. So, what do you mean by F satisfy property 3.3? What's the corresponding aggregation function for F?
>
> - This is a good question, and we have added the following clarification to the revised text below Equation 1.
> - The aggregation function $g$ is present in UMBC and is in fact be the sum that is within the cross attention layers and not a subsequent function.
> - For example, the cross attention between the slots/seeds and the input is given by $\sigma(SX^\top)$ which is then multiplied by $X$ (omitting the attention linear projections for simplicity).
> - If we are processing a single element at a time, then $\sigma(SX_i^\top) \in R^{k \times 1}$ and $X_i \in R^{1 \times d}$. The multiplication $\sigma(SX_i^\top)X_i$ is an outer product, and we can iteratively sum the outer products of vectors to form an outer product of matrices (i.e. $AA^\top = \sum_{i}\mathbf{a}_i\mathbf{a}_i^\top$).
> - Therefore, $g$ is in fact given by the sum in $\sum_{i=1}^N \sigma(SX_i^\top)X_i$.
>
> > Monte Carlo Slot Dropout: why does the composition of the functions and the dropout mask still a set function? Once the dropout mask is given, shouldn't the permutation of the slots affect the output?
>
> - The dropout mask on the set elements is applied directly between $f^*$ and $f$, and therefore is considered part of the input to $f^*$.
> - Slots permutations would be considered as part of the noise used to sample the slots (if they are random variables) and therefore any predictions are consistent given the noise used to sample the slots.
> - Predictions are therefore consistent given the dropout and slot noise.
>
> ---
>
> > Figure 4: It seems that UMBC + Diff. EM performs worse than Diff. EM alone, and this is also the case for UMBC + set transformer vs. Set transformer? I.e., UMBC + Diff. EM gives a larger negative log-likelihood, hence a lower log-likelihood.
>
> - This is correct, and expected. We highlight in Sections 4 and 5 as well as Appendix L that the UMBC model as described is effectively a bottleneck, albeit a bottleneck which imparts special properties into the entire composition, therefore a performance decrease may occur.
> - The experiments in Fig. 4 consider when the full set can be processed during test-time in order to evaluate the tradeoff for using gaining the MBC property.
> - In Figs. 2. and 4 show the same task, therefore if we require Set Transformer to accept a single point stream with no available memory for storage, then the performance would degrade rapidly (e.g. NLL $\approx 140$ in Fig. 2). Which model would you choose given the single point streaming constraint? (UMBC + Set Transformer is the best MBC guaranteed model with NLL $\approx 1.75$ from Fig. 4).
>
> ---
>
> > Table 1: Why does Oracle perform the worst in terms of NLL?
>
> - The Oracle in Table 1 is constructed by taking the empirical mean and covariance of the extracted features from the ImageNet training as the mixture parameters.
> -  In the empirical parameters, there may be noisy/uninformative dimensions which can be ignored by a deep model and which cause likelihoods to be far less than optimal.
> - Perhaps 'Oracle' was a misleading term, because it implies that it is the best case scenario.
> - Therefore, we have updated the text to refer to 'Empirical' rather than 'Oracle' in the revised version of the text.
>
> ---
> ### Thank you for your review, we remain open for further discussion.

---

### Official Review · Reviewer_BZo6 · 2022-10-27

**Confidence:** 3
**Correctness:** 2
**Technical Novelty And Significance:** 2
**Empirical Novelty And Significance:** 2
**Recommendation:** 3

**Clarity, Quality, Novelty And Reproducibility:**


**Clarity.**
In my opinion, the writing and clarity could be significantly improved.
For example, by reading the introduction, it is hard to understand what the motivation is, and it only becomes clearer in later sections. Please provide a specific example for a set function, and build on it to motivate focusing on MBC etc.
Some parts have very long sentences that are hard to understand, e.g., the paragraph on parallel UMBC Heads.

**Reproducibility**. The source code is provided.

### Questions
- Fig. 2: could you provide some intuition why Set Transformer performs that poorly here (given that in Fig. 4 it does well)?
- I do not understand why at the end of section 4 you have that dropout may give faster training. Note that dropout still does the same computation
- I am not sure what the last paragraph of section 4 means. Could you explain?

### Minor
- Lemma 4.1. It -> it;  and satisfies -> to  satisfy
- Fig. 2: it would be helpful to fix the $y$-axis to be the same range everywhere
- Table 3: seems like Set transformer should be in bold in the first column with the results

**Strength And Weaknesses:**

## Strengths
- This work seems to identify an important problem that having an input of varying size may be hard to handle for neural set functions
- the paper does a lot of empirical evaluations including comparison with baselines, dropout techniques, and ablation studies

## Weaknesses
- While this work seems to identify a relevant problem, I do not think it resolves it, please see the next part on motivation
- although there are numerous experiments, I do not see these as conclusive, that is, it is still unclear to me if the proposed solution is outperforming the baselines, and moreover, how it performs against naive implementations when the input is of variable length, see part below on experiments
- I find that the theoretical results are relatively simple arguments, see part below on theoretical results
- [minor] writing: see the section below



### Motivation
If I understand well, there are two main motivations to focus on this problem:
1. memory issue: when having varying input sizes, if we can set a maximum size of the input we can always use padding with zeros for example, or with some negative number if we use max-pooling etc. However, this naive strategy may result in very large memory requirements.
2. technical aspect: when we can not specify the maximum input size, the question that arises is how to technically deal with such inputs.

Regarding the former, this paper does not actually resolve the problem. See for example the part on self-attention starting from Proposition 4.1. To perform the normalization one needs to first do a pass to compute the individual normalization constants and then run in mini-batches. However, when backpropagating during training, the resulting model will be the full graph (as if the entire input was passed). Please describe how your method works on self-attention models, and provide wall clock and memory footprint comparisons with the baselines.

Regarding the latter, a baseline would be to simply pre-define a fixed input size and sample randomly that many samples. For example, if the input size is 1000, we can sample say pairs of elements, and we can fix to sample say 10000 such pairs (each time forming the pairs at random). This would increase significantly the dataset size, and the overall model (assuming the task is not simple to just do max-pooling) may perform well. Moreover, at inference time such a procedure would directly yield a confidence interval since multiple samples will be yielded from a full test input, thus the different predictions could be used for confidence intervals.
Please include such a baseline in your comparisons.

Lastly, I do not understand how, given an arbitrary aggregation function (not necessarily MBC), the herein proposed framework specifies that one proceeds  (as the abstract alludes to)? I understand that if that aggregation function is MBC then the entire map will be MBC as well. Nonetheless, often in practice, one does not know what is a suitable aggregation function (and standard ones may not be).  Hence I do not understand the claim of how *any* function can be transformed into an MBC map -- e.g. if there is prior knowledge that standard MBC functions are not suitable for the problem, but some other non-MBC one is, how could one proceed?


### Experiments

It seems like UMBC does not always improve upon the baseline, e.g. Diff. EM + UMBC in Fig. 4. Similarly, in Tab. 3 sometimes it does not provide improvement.
Could you provide more insights into this, e.g. additional datasets, or running UMBC with Deep Sets and Slot Set Enc?

Please also provide comparisons with a naive approach to solve this problem, see the part above on motivation.

Regarding the MCDropout, a natural question that arises is if the baseline methods significantly improve when it is used. Could you provide such experiments for completeness?


### Theoretical results
I find that Lemma 4.1. is relatively obvious since it is easy to see that if we aggregate the results in a way that satisfies the MBC property, the overall output will not change with respect to if we were to pass the full input).
I do not understand why Theorem 4.1. is not a simple corollary of Lemma 4.1, that is, since in Lemma 4.1  $f$ and $f^\star$ are arbitrary maps, the statement in theorem 4.1 follows immediately as a special case of it (otherwise, a proof of it would be needed).




**Summary Of The Paper:**

**Background**. This work focuses on the problem of so-called *neural set* functions, where the input is a set, the task could be clustering, classification etc., and the map from a set to an output is a neural network.
Since the input is a set, there are certain invariances that are relevant for this case, e.g., the output should not change if the order changes, etc.
Among these, most relevant to this work is the so-called *Mini-Batch Consistency* (MBC) property, which is relevant for the case when we cannot pass the entire input to the model and we have to partition the input.
The MBC property requires that the final output of an auxiliary function that we use should be the same as if we passed the entire (non-partitioned) input.
For example, consider we want to take the max element of a large vector: we may partition the input, apply max, and then apply max on top of that, which gives overall an MBC map.

**Summary**.
Let us say that the overall map is a composition of two functions, say feature and prediction part, where the input of the latter is the output of the former. This paper points out that it suffices that the feature map is an MBC map so that the overall map is MBC (and the prediction map can be an arbitrary function). This insight simplifies the architectural constraints of the prior work of Bruno et al (2020), and allows for handling inputs of arbitrary and varying size/dimension.
In addition, dropout is considered on the output of the feature function, as well as several comparisons with some existing prior works.





**Summary Of The Review:**

The paper points out a relevant abstract problem of neural set functions. It appears to me that the provided solution does not fully address the problem and the empirical results are insufficient (inconsistent benefit of the herein proposed UBMC, and lack of comparison with the simplest approach to address this).
This paper also considers dropout approaches on some parts of the model, but I do not think that these contribute to its novelty, since MCDropout is a general approach, especially since here it is applied to the remaining standard part of the neural net (after the aggregation).


My recommendation is only temporary, and I would be happy to raise my score if the authors address the major concerns or explain if I misunderstood.

---

> ### Author Response · Authors · 2022-11-14
> **Author Response (2/2)**
>
> > I do not understand how, given an arbitrary aggregation function (not necessarily MBC), the herein proposed framework specifies that one proceeds (as the abstract alludes to)? I do not understand the claim of how any function can be transformed into an MBC map -- e.g. if there is prior knowledge that standard MBC functions are not suitable for the problem, but some other non-MBC one is, how could one proceed?
>
> - This is the point we highlight in Figs. 2 and 4. Deepsets and SSE (MBC models) do not perform well for the task of Gaussian clustering (See Fig. 4). We know from prior works that Set Transformer [1] and Diff. EM. [2] (both not-MBC) perform well on this task. **We proceed by utilizing Thm 4.1 with the Set Transformer, creating an MBC composition which is more powerful than previous MBC models.**
> - We must stress the fact that **Fig. 4 shows that the best performing *MBC* models are both UMBC+Diff.EM. and UMBC+SetTransformer**, which are both MBC compositions of MBC+non-MBC models.
>
> ---
>
> > It seems like UMBC does not always improve upon the baseline, e.g. Diff. EM + UMBC in Fig. 4. Similarly, in Tab. 3 sometimes it does not provide improvement. Could you provide more insights into this
>
> - This is correct, and expected. We highlight in Sections 4 and 5 as well as Appendix L, the UMBC model as described is effectively a bottleneck, albeit a bottleneck with desirable properties, therefore a performance decrease *may* occur.
> - It is important to remember that UMBC does not aim to make non-MBC models generalize better in all scenarios, but instead UMBC gives them the ability to process partitioned sets in a consistent way.
>
> ---
>
> > Regarding the MCDropout, a natural question that arises is if the baseline methods significantly improve when it is used.
>
> - We have no doubt that dropout training/testing on the baselines will improve with the MCDropout method, as the results transitively apply to $f^*$.
> - However, this comparison misses the main advantage of UMBC. For example, **if we apply this type of dropout to Deepsets, then Deepsets would in fact cease to be MBC**, as one cannot perform MCDropout on the set (multiple forwards on the full set with a random selections of set elements) without requiring the whole set in memory. Therefore, it is the MBC projection to a cardinality of $K$ which allows for MCDropout inference to work while still satisfying MBC
> - In fact, the only way for Deepsets to accomplish this and maintain MBC, is to apply Theorem 4.1 and use a base UMBC module and perform dropout on the resulting set size of $K$, which highlights why Theorem 4.1 is useful.
>
> ---
>
> > Fig. 2: could you provide some intuition why Set Transformer performs that poorly here (given that in Fig. 4 it does well)?
>
> Yes, this issue highlights why MBC is important. In Fig. 2, there is a streaming constraint that the function must abide by. It must sequentially process the input and and update the embedding with no capacity to store streamed points.
>
> Therefore, the Set Transformer in Fig. 2 achieved the good performance (Fig. 4) when it is allowed to see the whole set at test time, but it performs very poorly when it is constrained to see one chunk at a time (the size and composition of the chunks are different for the different streaming settings shown in Figs. 2, 8 and listed in Appendix B.1.
>
> ---
>
> > I do not understand why at the end of section 4 you have that dropout may give faster training. Note that dropout still does the same computation
>
> - Please see Fig. 18 in the appendix, which shows that as the dropout rate approaches 1, the processing time goes to zero.
> - Note that *we are dropping out set elements and not model parameters or activations*, which means that higher dropout rates actually correspond to smaller input sets for $f^*$ and therefore less data for the model to process.
>
> ---
>
> > I am not sure what the last paragraph of section 4 means. Could you explain?
>
> - This paragraph states that it is possible to train $L$ independent parallel UMBC modules, with each one projecting a set from cardinality $N \mapsto K$)
> - If we then concatenate the resulting sets to form a set of cardinality $LK$, we can provide an input of cardinality $LK$ to $f^*$.
> - This allows the set to go through independent parallel attention layers.
> - We have updated this paragraph in the revised text.
>
> ---
>
> > Table 3: seems like Set transformer should be in bold in the first column
>
> That is correct, this was a typo and has been fixed.
>
> ---
>
> ### Thank you for your review, we remain open for further discussion.
>
> ### References
>
> [1] Lee, J., Lee, Y., Kim, J., Kosiorek, A., Choi, S., & Teh, Y. W. (2019, May). Set transformer: A framework for attention-based permutation-invariant neural networks. In International conference on machine learning (pp. 3744-3753). PMLR.
>
> [2] Kim, M. (2021, September). Differentiable Expectation-Maximization for Set Representation Learning. In International Conference on Learning Representations.

---

> ### Author Response · Authors · 2022-11-14
> **Author Response (1/2)**
>
> ### Thank you for taking the time to review our work, we will address each of your points in turn below.
>
> ---
>
> > If I understand well, there are two main motivations to focus on this problem…
>
> - We think there has been a general misunderstanding of the problem we are trying to solve. Firstly, there is no fixed input size, as a general set function should handle a set of any cardinality. The only reason for zero-padding which we can see would be to handle training batches with sets of different sizes, which is not an issue we consider, and an issue which already has solutions such as zero-padding as you mentioned.
> - The issue we consider is one where the test time set size is unknown and possibly large, and/or there are tight memory constraints which prevent the encoder from being able to store the entire set in memory, and processing must be done in pieces.
>
> ---
>
> > the part on self-attention starting from Proposition 4.1. To perform the normalization one needs to first do a pass to compute the individual normalization constants and then run in mini-batches.
>
> - This is incorrect. Proposition 4.1, and the UMBC formulation we consider, uses cross attention with learned parameters, not self-attention.
> - These parameters are introduced below Property 3.3 (denoted by $S$) and used in Equations 1-3.
> - The normalization constant can be incrementally updated at the arrival of a set element, which means that once we see an element of a set, we can process it, update the encoding and cumulative normalization constant, and never see it again, as described in the RHS of Equation 2.
>
> ---
>
> >However, when backpropagating during training, the resulting model will be the full graph (as if the entire input was passed).
>
> We think this is a misunderstanding. During training, UMBC (and also SSE) are limited to train-time set sizes for which the entire computation graph can fit into memory. Test time set sizes, however, become unlimited for UMBC models as no gradient is required, and memory no longer scales with the set size (*only for MBC models*) and the set can be processed in mini-batches and be guaranteed to achieve the same output. Handling larger-than-memory set sizes during training is an open problem and we look forward to seeing future research in this area.
>
> We have clarified this point in the first contribution bullet point in the introduction as well as the "Limitations and Future Work" in the appendix.
>
> ---
>
> >Please describe how your method works on self-attention models.
>
> For self attention, UMBC projects a set of any cardinality to one that is of cardinality $K$ (a constant). Therefore any subsequent self-attention blocks will receive a set size (sequence length) of $K$ which is independent of the original set size and incurs a constant memory overhead.
>
> ---
>
> > Please also provide comparisons with a naive approach to solve this problem, see the part above on motivation,
>
> > Moreover, at inference time such a procedure would directly yield a confidence interval since multiple samples will be yielded from a full test input, thus the different predictions could be used for confidence intervals. Please include such a baseline in your comparisons.
>
> This is possible, and our experiment in Fig. 2(b) does something very similar to this. We streamed random, unbiased chunks to the set transformer and averaged over all the predictions, which yielded the result. Additionally, Fig. 1 shows the variance in the encoded vector for different chunk sizes which shows that UMBC has no variance in the encoded vector, while the set transformer has a variance which depends on chunk size.
>
> - Please see the animation of the streams included in the supplementary file or link in the paper to see examples of why our example shows that a confidence interval will not work.
> - **Our experiments from Fig. 2 and 8, and the animated gif's in the paper link/supplementary file show that a confidence interval will still fail for non-MBC models**

---

### Official Review · Reviewer_cnQu · 2022-10-27

**Confidence:** 3
**Correctness:** 3
**Technical Novelty And Significance:** 2
**Empirical Novelty And Significance:** 2
**Recommendation:** 5

**Clarity, Quality, Novelty And Reproducibility:**

**Clarity.**  The paper is clearly written.

**Quality.**  Everything seems to be correct.

**Novelty.**  This paper has limited novelty in my opinion.  The theory seems to hold as a sole result of the definition of MBC, and the empirical results seem inconclusive.  So on the theoretical and empirical side, it's not clear what the novelty is here.

**Reproducibility.**  This seems reproducible.  It looks like the authors will release the code.

**Strength And Weaknesses:**

## Strengths

**Writing.**  The paper is well-written.  Of my batch of 7 papers, I would rank this as the second strongest in terms of writing.  The ideas are clearly articulated, and the flow of the paper makes sense.

**Experiments in different settings.**  This paper conducts a range of experiments, from clustering to ablation studies to distribution shift robustness.  This is commendable, and a strong point of the paper.

## Weaknesses

**Defining notation and key terms.**  There were a few points in the paper where I had trouble parsing terms introduced by the authors.  For example, the authors talk about consistency:

> "Subsequent work has highlighted the utility of Mini-Batch Consistency (MBC), the ability to sequentially process any permutation of a set partition scheme (e.g. streaming chunks of data) while maintaining consistency guarantees on the output."

At this point in the paper (only a few lines in), the reader has no way of knowing what "consistency" means here.  It becomes somewhat inferable by the time the reader gets to Section 3, but I fear that some readers will be lost.  It would be worth explaining the main ideas of the paper in a way that does not require technical definitions in the abstract.

Other feedback regarding similar weaknesses:

* The authors seem to use MBC to refer to the term "mini-batch consistency" in the abstract and "mini-batch consistent" in the intro.  It would be clearer if the authors stuck to using MBC to refer to one or the other of these two phrases.

* It's unclear what the authors mean by a "valid" set in Section 3.

* The presentation of set functions is confusing.  At the beginning of Section 3, set functions $f$ are said to map input sets $X$ to output sets $Y$.  However, later on, Property 3.1 tells us that set2vector functions map sets to one or several vectors.  And from this point onward, the authors seem to assume that all set functions are set2vector (e.g., in Thm. 4.1 -- the main result in this paper).  Given this, I think there needs to be clarification regarding what the input and outputs to the set functions are.  If we assume that $f$ maps sets to sets, then Thm. 4.1 seems to not apply.  If we assume that $f$ maps sets to vectors, then $g\circ f$ is vector valued, and therefore the composition $f^\star \circ g\circ f$ doesn't make sense if $f^\star$ takes sets (not vectors) as input.

* The notation in Property 3.2 is confusing.  The equation $f([x_{\pi(i)}, \dots x_{\pi(N)}]) = [f_{\pi(1)}(X_1), \dots, f_{\pi(n)}(x_n)]$ doesn't make sense to me.  Why does the permutation on the LHS go from $\pi(i)$ to $\pi(N)$ and why does the permutation on the RHS go from $\pi(1)$ to $\pi(n)$.  Should we assume that $N=n$?  What is $i$?

* What do the authors mean when they say that MBC "[added] a new dimension to the original view of Property 3.1?"  In what way does this add a dimension?  Are we to interpret this mathematically or intuitively?  If the former, how does dimension come into play?  If the latter, it's not clear *how* this adds dimension, and it would be helpful if the authors could expand on this point.

* Sets are often written as *elements* of $R^k$ for some $k$ or of $R^{n\times k}$ for some $n$ and $k$.  I find this relatively confusing, because there are also various vectors in play which are truly elements of Euclidean space.  Is there a better way to denote which objects are sets and which are vectors?  For example, is it correct to say that $\{1,2,3\}\in\R^3$?  I would argue that the answer is no, because $\{1,2,3\}$ denotes the same set as $\{3,2,1\}$, but clearly as vectors $[3,2,1]$ is not the same as $[1,2,3]$.  So there is an identifiably issue.

* The function Attention$(S,X)$ is used almost a full page before it is defined (on page 4).  This confused me as I was reading.

* In the definition of Property 3.3, what are $n_i$, $d$, and $d'$?  Should they be inferable from context?

* What is a slot-normalized activation function?  In general, after reading the paper, it was not clear to me what a "slot" is.  As this seems relatively important, it would be helpful if the authors could give intuition here.  One could certainly read the paper of Bruno et al., but for this paper to be self-contained and so as not to confuse readers, giving intuition for the most relevant related works seems important here.

* What is $\hat{\sigma}$, i.e., what does the hat denote?

* What do the authors mean when they say "arbitrarily hard MBC constraints."  What makes one constraint harder than another?  In what sense can this get "arbitrarily hard?"

* Why are MBC models like Deep Sets not able to leverage pairwise relationships?

* What do the authors mean by "simple" $f^\star$s?

* In the intro, the authors mention infinite set sizes.  So it's reasonable to ask: How does all of this extend to settings where the cardinality of the input is infinite (countably? uncountably?)?  How does one even construct a partition?

**Experiments.**  The experiments do not seem to support the conclusions made by the authors.  Indeed, I'm concerned that I missed something fundamental here, and if so, I hope that the authors will explain further.  However, it seems to me that baselines like Deep Sets ofter outperform the UMBC approach outlined by the authors.  For example, in Figure 4, the authors say that panels (e) and (f) show the best performance.  However, Diff. Em and Set Transformer (panels (c) and (d)) seem to reach lower values of the NLL for larger test set sizes.  Even for small test set sizes, (c) and (d) seem to do better.  Similarly, in Table 3, Deep Sets gets higher accuracy than all of the other methods on test sets of size 1000 and 2048.  Somewhat confusingly, these numbers are not bolded, despite the fact that (i) they perform better and (ii) (from my understanding) Deep Sets *is* MBC.  In the same categories, Deep Sets also outperforms UMBC on the NLL score.  And in Figure 6, it's not clear that UMBC does much better than the baselines.  So to summarize, the results seem relatively inconclusive as to why UMBC should be used when architectures like Deep Sets seem to offer strong performance.

**Theory.**  The theory is also not a particular strength of this paper.  It's relatively strong to call the results lemmas and theorem in my opinion.  Thm. 4.1 says that if you preprocess the data in an MBC way, any function will be MBC which seems self-evident.  That is, if I have a set function $f^\star$, if the input is in chunks, I can just use a separate MBC architecture $(g,f)$ to ensure that the output of the $g(f(X))$ is the same regardless of whether I chunked or not.  This holds directly by definition of MBC.

More broadly, the question is: What is the impact of this theory?  Does it change our understanding or result in significantly better empirical results?  And based on my understanding of the experiments (see the discussion above), I'm not sure that it does.

**Summary Of The Paper:**

This paper considers the problem of training neural networks which operate on sets.  The main focus is on the mini-batch consistency (MBC) property, which roughly states the following: Let $X$ be a set, and $(X_1, \dots, X_n)$.  Given a map $f$ from sets to vectors and an aggregation function $g$, the pair $(f,g)$ are MBC if $g(f(X_1), \dots, f(X_n)) = f(X)$.  Informally, this means that the aggregated output from processing the partition of $X$ should give the same output as if $X$ was passed directly through $f$.

The authors show that non-MBC functions can be made MBC by pre-processing the data with an MBC function.  They then introduce a so-called "universal" MBC (UMBC) function.  They then present several experiments using their UMBC.

**Summary Of The Review:**

To summarize, this paper is well-written and it has a broad array of experiments.  However, there are quite a few points of confusion in the notation and description of the setting.  Furthermore, I would argue that empirical results are inconclusive and that the theory does not constitute a significant contribution.  Therefore, I recommend that this paper not be accepted.

**Post-rebuttal.**  The authors fixed a number of typos, clarified the notation, reworded some parts of the text, and re-captioned the plots of some of the experiments.  All of this has improved the paper, and therefore I will raise my score from 3 --> 5.  However, as discussed in my response, I think that there are still some fundamental issues which make me lean toward reject.

---

> ### Author Response · Authors · 2022-11-14
> **Author Response (3/3)**
>
> >What is $\hat{\sigma}$ i.e., what does the hat denote?
>
> In the original text $\hat{\sigma}$ was defined as a softmax in the preceeding line. In the revision, however, the notation has been revised so that hat signifies normalization of the attention matrix.
>
> ---
>
> >What do the authors mean when they say "arbitrarily hard MBC constraints." What makes one constraint harder than another? In what sense can this get "arbitrarily hard?"
>
> - An example of this can be seen in the streaming settings which are included in Fig. 2 and 8 (for better visualization, please see animated gifs included in the supplementary file).
> - If the Set Transformer is forced to accept single point streams, its predictions becomes nearly useless (NLL $\approx 140$ in Fig. 2(a)). If however, it is given unbiased random sample of small subsets, it can make poor, but somewhat understandable predictions (NLL $\approx 16$ in Fig. 2(b)).
> - Therefore, constraints which dictate smaller chunk sizes and no storage capacity become increasingly harder for non-MBC models.
>
> ---
>
> >Why are MBC models like Deep Sets not able to leverage pairwise relationships?
>
> - In the feature extractors of Deepsets, each set element is processed by an independent row-wise operation, whereas self-attention or clustering as in Set Transformer and Diff. EM. both use pairwise information between set elements instead of independently processing each element.
> - For example, the clustering tasks require pairwise relationships between data.
> - This is why Set Transformer and Diff EM. are able to outperform DeepSets in Fig. 4
> - **Therefore, as UMBC can use Set Transformer as the $f^\ast$ model, UMBC+Set Transformer forms the best MBC model in Fig. 4**
>
> ---
>
> >What do the authors mean by "simple" $f^*$'s?
>
> See the answer above. In this sentence, simple $f^\ast$'s refer to $f^\ast$'s which consider simple row-wise processing of each set element before pooling (like a simple linear layer). We have added this clarification to the updated text in the "Amortized Clustering" paragraph on page 4.
>
> ---
>
> ### Thank you for your review, we remain open for further discussion.
>
> ### References
>
> [1] Andreis, B., Willette, J., Lee, J., & Hwang, S. J. (2021). Mini-Batch Consistent Slot Set Encoder for Scalable Set Encoding. arXiv preprint arXiv:2103.01615.
>
> [2] Lee, J., Lee, Y., Kim, J., Kosiorek, A., Choi, S., & Teh, Y. W. (2019, May). Set transformer: A framework for attention-based permutation-invariant neural networks. In International conference on machine learning (pp. 3744-3753). PMLR.

---

> > ### Comment · Reviewer_cnQu · 2022-11-17
> > **Thanks for your rebuttal; further comments**
> >
> > Thanks for the clarifications on the experiments.  It wasn't clear to me while reading that Fig. 2 and Fig. 4 showed the same experiment.  Perhaps rewording the sentence:
> >
> > > "We show the effect of different train/test set sizes in Figure 4."
> >
> > could help to clarify this.
> >
> > I'm still unsure of what is meant by the word "constraint" in this context.  Consider this sentence
> >
> > > "Therefore, faced with the constraint of Fig 2. which model would you choose?"
> >
> > Am I right in interpreting this as saying: "Given that we are going to be chunking the data and streaming it into the model piece-by-piece, which model achieves the best performance?"  In this respect, it might be helpful to demonstrate in Figure 2 what the gold standard is -- i.e., show what happens when you can pass the entire set through a non-MBC algorithm.  Then, in parts (c) and (d), it would be more clear that whereas the set transformer fails, the proposed UMBC methods perform quite well.  Obviously we shouldn't necessarily expect the UMBC algorithm to recover the same performance level as one would expect in the non-streaming setting, although in this case of Gaussian clustering maybe it's possible.
> >
> > All that being said, I agree that Figure 4 shows what one would hope to see, and I think that the new labeling of the Figures is helpful.  In this respect, it would also be helpful to add a couple of lines describing the table above Figure 1 (which I think should also be labeled as Table 1).  I missed this table the first time I looked through the paper, but it seems to be crucially important.  I also think that there may be a better way of displaying this data.  Perhaps having one figure for MBC algorithms and another figure for non-MBC algorithms would do the trick?  Ultimately, what we want to see clearly here is that there is a gap between MBC and non-MBC algorithms, and hopefully this gap is not too big.
> >
> > WRT to your comment: "Table 3 states that the bolded entries are between MBC/non-MBC model pairs," this was not my understanding based on how the paper is written.  This is what the caption says:
> >
> > > "**bold** entries denote the best performance between models with and without UMBC."
> >
> > I think it's reasonable to interpret this as meaning that the bolded entires compare *any* MBC algorithm with *any* non-MBC algorithm.  Why?  Because from a practical perspective, this is the comparison that we care about.  And what Table 3 seems to say is that for larger test set sizes, a practitioner should use Deep Sets based on the accuracy and NLL metrics, given that Deep Sets is also MBC.
> >
> > WRT to this comment:
> >
> > > "In Fig. 6, the 'Test' category shows the same results as the last column in Table 3. Therefore a has been observed. Also, throughout most of the cases, the UMBC+Set Transformer has and edge over other models at all corruption levels, and performs much better than the plain Set Transformer."
> >
> > I still do not understand.  Looking at Figure 6, it's unclear which method I would prefer.  For instance, at corruption level 5, when I compare the quantiles of Deep Sets to the quantiles of UMBC + Set Transformer, it's unclear why I should prefer UMBC + Set Transformer.  This holds true mostly for all of the corruption intensities.  I do agree that adding UMBC + [algorithm] (where [algorithm] == either Set Transformer or Diff. EM) tends to improve over [algorithm] without UMBC, but again, it seems more relevant to compare this to Deep Sets and SSE.
> >
> > This nuance is at the core of the question: What is the impact of the theory?  It seems that in some cases, UMBC + [algorithm] improves on plain [algorithm], but doesn't improve over the standard Deep Sets baseline.  In other cases, adding UMBC does seem to offer an improvement over SSE and Deep Sets.  So to make a strong case for why this theory is impactful, it seems important to find the conditions under which UMBC + [algorithm] results in a new procedure which does better than any possible MBC baseline.
> >
> > I think that the remaining changes have improved the paper, including cleaning up the notation and adding some clarifications.  All in all, post rebuttal I think that the paper has improved, and therefore I will increase my score.  However, I do still think that there are lingering issues (as discussed above) which are somewhat fundamental, in the sense that it is not clear when this algorithm actually improves over the MBC baselines.

---

> > > ### Author Response · Authors · 2022-11-18
> > > **Further Discussion and Clarifications**
> > >
> > > ### Thank you for the discussion, please see below:
> > >
> > > ---
> > >
> > > > This nuance is at the core of the question: What is the impact of the theory? It seems that in some cases, UMBC + [algorithm] improves on plain [algorithm], but doesn't improve over the standard Deep Sets baseline. In other cases, adding UMBC does seem to offer an improvement over SSE and Deep Sets.
> > >
> > > - **This is precisely why it is important to have a method of achieving a _Universal MBC function_ which gives practitioners the freedom to choose from a wider variety of functions**.
> > > - The simple fact is, there will be some tasks where existing MBC models perform well, and there will be tasks where one wishes they had could access a more diverse set of functions.
> > > - For example:
> > >   1. If faced with a problem such as MBC amortized clustering of sets, one would know from both prior work and intuition that a self-attentive or clustering based model would likely be the best choice.
> > >   2. But, prior to our work, one would be limited to either:
> > >      - Currently known MBC arcitectures such as DeepSets/SSE, OR...
> > >      - Be forced to accept poor, inconsistent MBC performance from non-MBC models as demonstrated in Figs. 1, 2, and 8.
> > >
> > > > So to make a strong case for why this theory is impactful, it seems important to find the conditions under which UMBC + [algorithm] results in a new procedure which does better than any possible MBC baseline.
> > >
> > > **Fig. 4 shows what you seek. UMBC+[non-MBC algorithm] performs better than any possible MBC baseline.**
> > >
> > > ---
> > >
> > > > In this respect, [...] demonstrate in Figure 2 what the gold standard is -- i.e., show what happens when you can pass the entire set through a non-MBC algorithm. Then, in parts (c) and (d), it would be more clear that whereas the set transformer fails, the proposed UMBC methods perform quite well.
> > >
> > > - **This is precisely what we convey in Fig. 4. which shows what the gold standard is in terms of NLL.**
> > > - In Fig. 4, the Set Transformer performs slightly better than UMBC+Set Transformer, which is expected.
> > >
> > > ---
> > >
> > > > I think it's reasonable to interpret this as meaning that the bolded entires compare any MBC algorithm with any non-MBC algorithm. Why? Because from a practical perspective, this is the comparison that we care about. [...] Table 3 seems to say that for larger test sets, a practitioner should use Deep Sets based on the [...] metrics, given that Deep Sets is also MBC.
> > >
> > > - We have added another line to the caption stating that the top row is present for reference.
> > > - Again, **the point of Table 3 is to show the tradeoff for using MBC**.
> > > - A practitioner indeed probably should choose Deepsets for a task like ModelNet40.
> > >   - **The same practitioner faced with a clustering task, however, would benefit from Theorem 4.1, and the UMBC composition.**
> > >   - **The same practitioner could also use the insights from Table 3 to gain more confidence about what is being traded for using a UMBC composition.**
> > > - Our goal and focus of this paper is not to make the absolute best generalizing Set Model, it is to show that we can impart the MBC property in *ALL* set models by using a UMBC composition.
> > >
> > > ---
> > >
> > > > Looking at Figure 6, it's unclear which method I would prefer. [...] when I compare the quantiles of Deep Sets to the quantiles of UMBC + Set Transformer, it's unclear why I should prefer UMBC + Set Transformer. [...]. I do agree that adding UMBC + [algorithm] (where [algorithm] == either Set Transformer or Diff. EM) tends to improve over [algorithm] without UMBC, [...] it's more relevant to compare this to Deep Sets/SSE.
> > >
> > > - Again, our goal is not to make the absolute best generalizing set model, **our goal is to impart the MBC property universally in all set functions and analyze the effects of doing so**.
> > > - Figure 6 as well should be viewed in terms of how it affects the underlying model, and as you point out, there are positive effects.
> > >
> > > ---
> > >
> > > ### Revisions
> > >
> > > > It wasn't clear to me while reading that Fig. 2 and Fig. 4 showed the same experiment. Perhaps rewording the sentence
> > >
> > > - We have added it to the caption of Fig. 4 in the latest revision of the text.
> > > - We would like to point out that the original text stated this in the "Amortized Clustering" section of the experiments.
> > >
> > > > Am I right in interpreting this as saying: "Given that we are going to be chunking the data and streaming it into the model piece-by-piece, which model achieves the best performance?"
> > >
> > > Yes, this is correct.
> > >
> > > > [...] add a couple of lines describing the table above Figure 1 (which I think should also be labeled as Table 1). I missed this table the first time I looked through the paper, but it seems to be crucially important.
> > >
> > > We have moved the table to be independent with its own caption.
> > >
> > > > could help to clarify this.
> > >
> > > This sentence near the bottom of page 6 has been reworded.
> > >
> > > ---
> > >
> > > ### Thank you for the discussion, we have done our best to incorporate the suggested changes, and remain open to discussion until the end of the disc. period.

---

> ### Author Response · Authors · 2022-11-14
> **Author Response (2/3)**
>
> >How does all of this extend to settings where the cardinality of the input is infinite (countably? uncountably?)? How does one even construct a partition?
>
> This is an important question, and highlights the utility of our theory. With UMBC, we simply don't care if the set is countable or how to partition, because Theorem 4.1 frees us from this burden. For example, consider the animated gif's included at the link in the paper and in the supplementary file. Assume that all the streams now become infinite at test time. UMBC+Set Transformer will continue accepting new points forever with a constant memory overhead, while the Set Transformer will have an infinitely expanding memory overhead and will have to face the decision of storing the streamed points, then selecting which ones to accept, which ones to discard, how and if they should be partitioned, etc.
>
> ---
>
> >The presentation of set functions is confusing. At the beginning of Section 3, set functions  are said to map input sets $X$ to output sets $Y$. However, later on, Property 3.1 tells us that S2V functions map sets to one or several vectors. And from this point onward, the authors seem to assume that all set functions are set2vector (e.g., in Thm. 4.1 -- the main result in this paper). Given this, I think there needs to be clarification regarding what the input and outputs to the set functions are. If we assume that $f$ maps sets to sets, then Thm. 4.1 seems to not apply. If we assume that $f$ maps sets to vectors, then $g \circ f$  is vector valued, and therefore the composition $f^* \circ g \circ f$ doesn't make sense if $f^*$ takes sets (not vectors) as input.
>
> >Sets are often written as elements of $R^k$ for some $k$ or of $R^{n \times k}$ for some $n$ and $k$. I find this relatively confusing.
>
> - This is a good point, and we have clarified this in the revised version of our paper at the beginning of section 4. The insight is that a S2V function which has an output $Z \in \mathbb{R}^{K \times d}$ can be re-interpreted as a set consisting of $K$ elements with each element $\mathbf{x}_i \in \mathbb{R}^{d}$. Therefore $f^*$ is receiving a set valued input.
>
> - For completeness, we must point out that $f^*$ is a set function, and therefore Thm. 4.1 applies regardless of whether the input is a single vector (set cardinality $1$) or a set (cardinality $N$).
>
> - Theorem 4.1 has been rewritten for clarity in the updated version of the text.
> ---
>
> >Authors use "mini-batch consistency," and "mini-batch consistent." It would be clearer to just use MBC.
>
> We see your point and have changed to use MBC after the first introduction of the word in most instances.
>
> ---
>
> >It's unclear what the authors mean by a "valid" set in Section 3.
>
> The word valid has been changed to *possible*
>
> ---
>
> >The notation in Property 3.2 is confusing (permutations go from $(i,N)$ and $(1,n)$...
>
> This is a typo and has been fixed. Both sides of the equation should start at $1$ and end at $N$.
>
> ---
>
> > What do the authors mean when they say that MBC "[added] a new dimension to the original view of Property 3.1?"
>
> Dimension was meant to intuitively, but we revised the sentence to be more precise. If we consider each set element $x_i$ to have another dimension $p$ added to the indices $x_{i,p}$, where $p$ represents the the random partition index, then an MBC function must be invariant to all
> permutations $\pi$ of the indices $(\pi(i),\pi^\prime(p))$.
>
> ---
>
> >The function Attention(S, X) is used almost a full page before it is defined (on page 4). This confused me as I was reading.
>
> We have added another equivalence which briefly defines self-attention and references the original work.
>
> ---
>
> > In the definition of Property 3.3, what are $n_i, d$, and $d^\prime$? Should they be inferable from context?
>
> $d^\prime$ is listed as the projection dimension of $f$. The $n_i$ was supposed to represent a subset of indices of $N$ as we inferred from the original notation from [1]. We agree this can be clearer, so we have updated Property 3.3 and Definition 3.1.
>
> (latex would not render properly in this comment, please see the revised text)
>
> ---
>
> >What is a slot-normalized activation function? In general, after reading the paper, it was not clear to me what a "slot" is.
>
> We have updated the text after Property 3.3 to define what a slot normalized activation function is. Essentially, a traditional dot product attention matrix $A$ is normalized over the rows by the operation $A = \text{softmax}(QK^\top, \text{dim=``rows"})$. Slots take the position of $Q$ in our formulation, so therefore slot-normalized activation function refers to one which normalizes over the columns  $A = \text{softmax}(QK^\top, \text{dim=`columns'})$.
>
> Additionally, slots are learnable parameters which take the place of $Q$ in standard attention. These are defined in relation to attention after Property 3.3 and in Equation 1. For context, [1] calls them slots and [2] calls them seeds.

---

> ### Author Response · Authors · 2022-11-14
> **Author Response (1/3)**
>
> ### Thank you for taking the time to review our work, we will address each of your points in turn below.
>
> ---
>
> >The experiments do not seem to support the conclusions made by the authors. I'm concerned that I missed something fundamental here, and if so, I hope that the authors will explain further.
>
> >Figure 4, the authors say that panels (e) and (f) show the best performance. However, Diff. Em and Set Transformer (panels (c) and (d)) seem to reach lower values of the NLL for larger test set sizes.
>
> >So to summarize, the results seem relatively inconclusive as to why UMBC should be used when architectures like Deep Sets seem to offer strong performance.
>
> We think there has been a misunderstanding, which we will attempt to address here.
>
> - Figs. 2 and 4 show the same experiment
> - Fig. 2 demonstrates that any non-MBC model can be broken by constraining computational resources in a streaming scenario.
> - Therefore, faced with the constraint of Fig 2. which model would you choose?
>   - **Set Transformer gave $NLL \approx 140$ (Fig. 2) and UMBC+Set Transformer is the best MBC model with NLL $\approx 1.75$ (Fig. 4)** for streaming data in this experiment.
>   - **Therefore UMBC+Set Transformer and UMBC+Diff EM. (both MBC+non-MBC compositions) outperform existing MBC models.**
> - The first sentences in Section 5 state that our goal in the experiments is to evaluate the effect of the composition $f^*(\text{UMBC(.)})$. Therefore the experiments consider the baseline models when the full set is available in one chunk.
> - Fig 4. states that among MBC models (subfigures c, d, e, f) UMBC+Set Transformer performs the best.
> - We have added bolded and color coded sub-captions as well clarifying the MBC status of each model.
> - As we mention throughout the paper, UMBC is a bottleneck, albeit one with desirable properties (e.g. Thm. 4.1), and therefore a decrease in performance over the unaltered model may occur.
> - **Diff. EM and Set Transformer have been included in Fig. 4 for completeness to show what is being exchanged for gaining the MBC property, while Fig. 2 shows the extreme breakdown which may occur in an MBC setting**.
>
> ---
> >Table 3, Deep Sets gets higher accuracy than all of the other methods on test sets of size 1000 and 2048. Somewhat confusingly, these numbers are not bolded, despite the fact that (i) they perform better and (ii) (from my understanding) Deep Sets is MBC. In the same categories, Deep Sets also outperforms UMBC on the NLL score.
>
> Table 3 states that the bolded entries are between MBC/non-MBC model pairs, which is why the table has horizontal dividers. Again, this is meant to show that the bottleneck has minimal overall impact on performance while giving interesting new qualities which we desire, such as the MBC property and improved calibration and likelihood over the unaltered model.
>
> ---
>
> > Figure 6, it's not clear that UMBC does much better than the baselines.
>
> In Fig. 6, the 'Test' category shows the same results as the last column in Table 3. Therefore a large improvement in in-distribution ECE has been observed. Also, throughout most of the cases, the UMBC+Set Transformer has and edge over other models at all corruption levels, and performs much better than the plain Set Transformer.
>
> ---
>
> > Thm. 4.1 says that if you preprocess the data in an MBC way, any function will be MBC which seems self-evident. That is, if I have a set function $f^*$, if the input is in chunks, I can just use a separate MBC architecture $(g, f)$ to ensure that the output of the $g(f(X)$ is the same regardless of whether I chunked or not. This holds directly by definition of MBC.
>
> >More broadly, the question is: What is the impact of this theory? Does it change our understanding or result in significantly better empirical results? And based on my understanding of the experiments (see the discussion above), I'm not sure that it does.
>
> - Thm. 4.1 may seem self evident in hindsight, but we must point out that before deriving Thm. 4.1, such a composition of functions had never been considered nor evaluated.
> - As for the impact of the theory, **it does lead to significantly better empirical results. Faced with the same task as Fig. 2 and 4, and a constraint of any of the streaming settings (Fig. 2, 8 and the included animations), which model would you choose? The UMBC+Set Transformer give the best MBC gauranteed performance (NLL $\approx 1.75$)**

---

### Author Response · Authors · 2022-11-14
**Shared Response**

We thank the reviewers for taking the time to review our work. In addition to the individualized responses, we would like to include a shared response to all viewers to address some common themes.

---

*There seems to be a misunderstanding regarding Fig. 4 showing better performance for Set Transformer [1] and Diff. EM. [2]*


- Fig. 4 caption states that the best performing **MBC** models are UMBC+Diff EM. and UMBC+Set Transformer.
- Section 5 states that these experiments are performed when the full test set is available. This is contrary to Fig. 2 which shows what happens when Set Transformer is forced to accept streaming data with no storage capacity.
- We have added bolded and color coded captions in Fig. 4 to highlight this point. If there is anything we can do to make this clearer, we are happpy to make further changes.
- Note that Set Transformer gives NLL $\approx 140$ (Fig. 2) when forced to accept streaming inputs with no storage capacity, while UMBC+Set Transformer gives an MBC guaranteed NLL $\approx 1.75$ (Fig. 4).
- For a real-time example of what this looks like, see the animated gif's included in the link in the paper and the supplementary file.

---


*Multiple reviewers questions the performance of Table 4, which is not much higher than other baselines.*

- There are an infinite amount of hard streaming settings we could enforce on a non-MBC baseline to make it fail (some are demonstrated in Fig. 2, Fig. 8 and the animations included in the supplementary file.)
- Therefore, after demonstrating that the non-MBC baselines can be made to fail in a variety of settings, we evaluate the models in the full batch setting in order to show the performance tradeoff
- To show this, we evaluated the Set Transformer against the UMBC+Set Transformer with slot-softmax activation in the below table
- If we evaluate a streaming setting of different chunk sizes (column headings in the table below), we can see that **Set Transformer fails due to the fact that it is not-MBC.**
- **(NOTE: small fluctuations in UMBC numbers are due to floating point errors during aggregation)**


| Metric$\downarrow$/Chunk Size$\rightarrow$ |            Model            |   1   |   2   |   4   |   8   |   16  |
|------|---------------------------|:-----:|:-----:|:-----:|:-----:|:-----:|
|  ACC.  |       Set Transformer       |  5.69 |  5.33 |  6.30 | 10.39 |  21.9 |
|        | UMBC+Set Transformer (Ours) | **86.32** | **86.38** | **86.43** | **86.53** | **86.47** |
|  NLL.  |       Set Transformer       | 33.35 | 42.16 | 38.99 | 29.49 | 19.51 |
|        | UMBC+Set Transformer (Ours) |  **0.54** |  **0.54** |  **0.53** |  **0.53** |  **0.53** |
|  ECE  |       Set Transformer       | 81.65 | 83.43 | 86.24 | 82.24 | 69.53 |
|        | UMBC+Set Transformer (Ours) |  **3.01** |  **3.16** |  **3.10** |  **3.09** |  **3.16** |



---

*Why does Set Transformer perform poorly in Fig. 2, given that in Fig. 4 it does well?*

This reason for this highlights why MBC is important. In Fig. 2, there is a streaming constraint that the function must abide by. It must sequentially process the input and has no capacity to store streamed points.

Therefore, the Set Transformer achieved the good performance (Fig. 4) when it is allowed to see the whole set at test time, but it performs very poorly when it is constrained to see one chunk at a time (the size and composition of the chunks are different for the different streaming settings shown in Figs. 2, 8 and listed in Appendix B.1.

We have updated the caption of Fig. 2 to clarify this point in the revised text.

---

### Thank you for the discussion thus far, we remain open for further discussion until the end of the discussion period.

### References

[1] Lee, J., Lee, Y., Kim, J., Kosiorek, A., Choi, S., & Teh, Y. W. (2019, May). Set transformer: A framework for attention-based permutation-invariant neural networks. In International conference on machine learning (pp. 3744-3753). PMLR.

[2] Kim, M. (2021, September). Differentiable Expectation-Maximization for Set Representation Learning. In International Conference on Learning Representations.

---

### Author Response · Authors · 2022-11-25
**Have Your Concerns Been Addressed?**

### Dear Reviewers,

---

We have done our best to address each and every concern raised and we remain open to further discussion until the end of the discussion period. Please see our brief summary below:

---

### cnQu

In our continued discussion (https://openreview.net/forum?id=FWl6TFsE7Cp&noteId=ZH1eeFqWSz), we have highlighted the following:

- The utility of our method lies in the expansion of the known Set of MBC architectures, which provides more options to practitioners.
- Additionally, The instances where UMBC + [algorithm] results in the best UMBC performance (Fig. 4 and Table 2).
- Our empirical findings indicate that when the underlying [algorithm] has a natural edge over non-MBC models, then converting the model to be UMBC + [algorithm] preserves the performance characteristics while also imparting the MBC property.

---

### BZo6

In our response to your initial review, we have highlighted the following:

- We have tried to clear up what we think is a general misunderstanding about UMBC. The UMBC module itself cannot use self-attention, as this is not possible without seeing the full set at once. Instead it is the downstream set encoder which is allowed to use self-attention after the UMBC layer. Theorem 4.1 allows this second function to be an unconstrained set function.
- A similar approach to the naive baseline which you suggested has been tried in the streaming experiments present in Fig. 2. As can be seen in the animations included in the github link and supplementary file, the addition of a confidence interval will not provide any meaningful help to the naive baseline.

---

### 9w5z

In our response to your initial review, we have highlighted the following:

- The MBC aggregation function in our model is actually the summation which is present in the final attention multiplication within UMBC. This clarification was added to the revised version of the paper.
- Fig. 4 shows the Set Transformer for reference. The best performing MBC model for this task is UMBC+Set Transformer. (and Set Transformer can catastrophically fail in MBC test-settings as shown in Fig. 2)

---

### SNvt

In our response to your initial review, we have highlighted the following:

- The novelty in our approach lies in the application of Theorem 4.1 which greatly expands the possible set of MBC architectures which can be used.
- We have shown this to be most beneficial for tasks on which the non-MBC models excel (Fig. 4 and Table 2)

---

### We value the time and effort in reviewing our work. We remain open to further discussion, and we sincerely hope we have addressed all of your concerns.

### Thank you,
### Authors

---

### Decision · Program_Chairs · 2023-01-20

**Decision:**

Reject

**Justification For Why Not Higher Score:**

The reviewers were not convinced of the empirical significance of the results. Theoretically, the results were fairly straightforward, so also not enough to stand on their own.

**Justification For Why Not Lower Score:**

N/A

**Metareview: Summary, Strengths And Weaknesses:**

The paper addresses the MBC (mini-batch-consistency) property, introduced in a recent paper by Bruno et al '21, which roughly requires that the output of a trained function is invariant to the ordering of the data. Precisely, for any set $X$, and any partition of the set into subsets $(X_1, \dots, X_N)$, we want $g(f(X_1), \dots, f(X_N)) = f(X)$, where $g$ is an aggregation function.  On the theory end, the authors show that non-MBC functions can be made MBC, loosely by pre-processing with an MBC function (Theorem 4.1). They also introduce a Monte Carlo dropout strategy that seems to help with robustness.

The discussion revolved both around the technical contributions and the significance of the results. The main theoretical result (Theorem 4.1) is fairly straightforward (only a few lines). That on its own would not be a problem, though the reviewers were not convinced of the significance of the results. For example, both authors and reviewers agree that the framework provided makes it easier to design MBC functions. Overall, however, they were not convinced of the comparison to non-MBC baselines, and indirectly, of the value of having a very expressive MBC function as opposed to a non-MBC one.